# Prediction-Powered E-Values

**Daniel Csillag** [1]    **Claudio José Struchiner** [1]    **Guilherme Tegoni Goedert** [1]

## Abstract

Quality statistical inference requires a sufficient amount of data, which can be missing or hard to obtain. To this end, prediction-powered inference has risen as a promising methodology, but existing approaches are largely limited to Z-estimation problems such as inference of means and quantiles. In this paper, we apply ideas of prediction-powered inference to e-values. By doing so, we inherit all the usual benefits of e-values – such as anytime-validity, post-hoc validity and versatile sequential inference – as well as greatly expand the set of inferences achievable in a prediction-powered manner. In particular, we show that every inference procedure that can be framed in terms of e-values has a prediction-powered counterpart, given by our method. We showcase the effectiveness of our framework across a wide range of inference tasks, from simple hypothesis testing and confidence intervals to more involved procedures for change-point detection and causal discovery, which were out of reach of previous techniques. Our approach is modular and easily integrable into existing algorithms, making it a compelling choice for practical applications.

## 1. Introduction

Statistical inference is ubiquitous in many critical areas of application, such as medicine and economics. Central to their use is the availability of moderate amounts of data to empower our inferences. However, such data can be expensive to obtain, which complicates matters.

A common strategy is to simply collect a smaller amount of data, in order to minimize costs. Unfortunately, this generally leads to more uncertain inferences. Alternatively, there are methods that leverage auxiliary cheap-to-obtain data

to 'compensate' for the missing expensive data. Classical works in this direction include single imputation and multiple imputation methods (Little & Rubin, 2019), but they generally lack any strong guarantee of correctness. More recently, (Angelopoulos et al., 2023a) proposed prediction-powered inference, which allow for versatile procedures that benefit from strong correctness guarantees, notably including unbiasedness and type-I error control under very light assumptions.

At its heart, the idea of prediction-powered inference is simple: we leverage a predictive model (which can be arbitrarily complex, e.g., large neural networks) to predict the expensive data from the cheap data. We can then use our whole dataset to perform our inference by imputing missing expensive data with predictions from our model, while leveraging the available expensive data to quantify our model's inaccuracies, debiasing our inference.

Prediction-powered inference has already inspired a large amount of literature, both methodology-wise (e.g., (Zrnic & Candes, 2024; Angelopoulos et al., 2023b; Zrnic & Candès, 2023; Gu & Xia, 2024)), as well as in applications such as language model evaluations (Chatzi et al., 2024; Boyeau et al., 2024), genome-wide association studies (Miao et al., 2024) and more. However, throughout, the inference tasks considered are fairly limited; previous works are essentially restricted to problems that can be framed in terms of Z-estimation,[1] which includes many common tasks such as inference of means, quantiles and regression coefficients, but not much more. In this paper, we significantly expand this frontier by applying prediction-powered inference to e-values.

E-values are a recent enticing alternative to p-values. Formally, an e-value for a null hypothesis $H_0$ is a nonnegative real random variable $E$ such that, if $H_0$ holds, then $\mathbb{E}[E] \leq 1$; by Markov's inequality, it is then unlikely that the e-value $E$ is high under the null, and thus a high e-value ($\gg 1$) provides evidence against the null hypothesis. Though simple, this is a very powerful notion: e-values allow for powerful procedures under very lax assumptions (e.g., not even i.i.d., nonparametric and nonasymp-

---

*Equal contribution  [1]School of Applied Mathematics, Getulio Vargas Foundation, Rio de Janeiro, Brazil. Correspondence to: Daniel Csillag <daniel.csillag@fgv.br>.

*Proceedings of the 42$^{nd}$ International Conference on Machine Learning*, Vancouver, Canada. PMLR 267, 2025. Copyright 2025 by the author(s).

---

[1]A Z-estimation problem is one in which we seek to infer a parameter $\theta^\star \in \Theta$ such that $\mathbb{E}_Z[\psi(Z; \theta^\star)] = 0$, for some known function $\psi$.

totic) (Howard et al., 2018), naturally handle sequential and anytime-valid inference (Ramdas et al., 2022), naturally fit into multiple testing and post-selective inference (Wang & Ramdas, 2020; Xu et al., 2022) and allows for significance levels to be chosen a posteriori (Koning, 2023; Grünwald, 2022) – properties that are notoriously challenging to obtain with the more standard p-values, if not outright impossible, especially in conjunction. Furthermore, e-values are rather universal: any e-value can be converted to a p-value by simply taking its reciprocal, and any p-value can be converted to an e-value by a process termed calibration (Vovk & Wang, 2019), albeit at a slight loss of power.

By working atop e-values, our procedure gains a great amount of versatility. We show that **any inference procedure that operates in terms of e-values has a prediction-powered counterpart**, given by our method. Moreover, our procedure naturally inherits all of the usual virtues of e-values, in particular including anytime-validity and post-hoc validity. In fact, the sequential nature of our procedure further empowers prediction-powered inference methods, allowing us to arbitrarily improve our predictive model and data collection policy over the course of the inference, whereas previous methods require us to fix it a priori, or learn it from a separate data split.

We illustrate our procedure in four case studies. First, we use it for a simple problem of estimating prevalence of diabetes on a population from readily available survey data. Secondly, we apply our method for a problem of anytime-valid testing of the hypothesis that a deployed model's risk does not exceed a certain safety level, for the purpose of continuous risk monitoring. We then turn to more involved inference tasks. On the same context of continuous risk monitoring, we apply our method for detection of change-points, in which we seek to identify points in time where some aspect of the time series has changed. Finally, we consider how our method enables powerful procedures for causal discovery under missing (costly) data.

**Our contributions**

1. We present a new method for prediction-powered inference based on e-values. Besides being applicable to a much more general setting than the ones previously considered in the literature, it inherits all the usual benefits of e-values, including sequential inference that is valid under arbitrary optional stopping and post-hoc validity. Moreover, it allows for the underlying predictive model to be updated over the course of the inference, yielding much better data efficiency compared to prior work (which require the model to be fit on a separate data split);

2. We show how the base method can be extended from simple hypothesis testing with e-values to more in-

volved procedures, first considering confidence intervals/sequences and then general algorithms based on e-values. In particular, we show that simply substituting the base e-values by our prediction-powered e-values yields valid prediction-powered procedures that are statistically powerful, leading to a modular and widely applicable technique.

3. We showcase our method in four case studies ranging from simple mean estimation and hypothesis testing to change-point detection and causal discovery. This highlights the wide applicability of our approach, and we consistently note its much improved performance compared to baselines in spite of substantial (often 100x-200x) reductions in data acquisition costs.

**Related work** Our method, much like most of the prediction-powered inference literature, is fundamentally connected to the literature on semiparametric inference. In particular, our prediction-powered e-values can be seen as a use of the AIPW estimator (Robins et al., 1994), but atop the e-values rather than the data. Specifically connecting to e-values, prior work by (Xu et al., 2024) has explored using an AIPW estimator atop e-values for the mean in order to construct risk-controlling prediction sets (RCPSs); their 'variance-reduced' method turns out to be a special case of our approach.

## 2. A General Method

We will first present how we can transform a standard e-value into a prediction-powered one in the context of hypothesis testing. This mechanism can then be leveraged to transform more complex procedures powered by e-values into prediction-powered ones; we first thoroughly instantiate this for confidence sequences, and then more generally in the context of general e-value-powered algorithms. Throughout, we consider an active data collection setting.

### 2.1. Hypothesis testing

Our goal is to test some null hypothesis $H_0$, and for this purpose have a stream of data $(X_i, Y_i)_{i=1}^{\infty}$. The $X_*$ correspond to 'cheap' data that we will always have access to, while the $Y_*$ correspond to data that is expensive to obtain, and as such we have little access to – but, ultimately, the hypothesis we want to test is over the distribution of the $Y_*$.s

Data acquisition costs aside, a sound approach to perform such a hypothesis test is to leverage an e-value $E_n$ – i.e., a nonnegative random variable that is a function of the first $n$ data points, such that under the null $H_0$ it holds that $\mathbb{E}[E_n] \leq 1$. In particular, we consider e-values of the form

$$E_n := \prod_{i=1}^{n} e_i(Y_i), \tag{1}$$

where $(e_i)_{i=1}^\infty$ is a predictable sequence of the 'components' of the e-value, i.e., each $e_i$ can be arbitrarily dependent on the samples before time $i$ (but nothing else). We will further require that the e-value's components be predictably bounded: for all $i$, $e_i(\cdot) \in [a_i, b_i]$ for some predictable sequences $(a_i)_{i=1}^\infty$ and $(b_i)_{i=1}^\infty$, and with $a_i > 0$ for all $i$.

Most e-values in the literature are already of this form (e.g., (Waudby-Smith & Ramdas, 2020; Podkopaev & Ramdas, 2023a;b; Waudby-Smith et al., 2022; Bar et al., 2024)), or can factored into it. The boundedness assumption can be enforced by simple rescaling and clipping, albeit at a slight loss of power.

Should we have access to *perfect* models $\mu_i^\star : \mathcal{X} \to \mathbb{R}$, i.e., such that $\mu_i^\star(X_i) = Y_i$ almost surely, then we could instead only use the predictions atop the cheaper data, $\mu_i^\star(X_i)$, to construct the e-value by its components:

$$E_n^{\text{imputed}} := \prod_{i=1}^n e_i(\mu_i^\star(X_i)).$$

However, in the much more realistic scenario that the model is not perfect, $E_n^{\text{imputed}}$ will not be a valid e-value.

We can, however, debias $E_n^{\text{imputed}}$ as per prediction-powered inference (Angelopoulos et al., 2023a) and active statistical inference (Zrnic & Candes, 2024). First, endow the data stream with additional random variables $\xi_i \sim \text{Bern}(\pi_i(X_i))$ denoting whether we should collect (and thus have access) to the more expensive data $Y_i$, where $\pi_1, \pi_2, \ldots : \mathcal{X} \to [1 - a_i/b_i, 1]$ is a predictable (i.e., possibly arbitrarily dependent on data prior to $i$, but independent of all from $i$ onwards) sequence of functions that produce the probability of data collection.

With this augmented data stream $(X_i, Y_i, \pi_i, \xi_i)_{i=1}^\infty$, we can form a new 'prediction-powered' sequence of e-values, with form similar to that of the active prediction-powered estimators of (Zrnic & Candes, 2024):

$$e_i^{\text{ppi}} := e_i(\mu_i(X_i)) + \left[e_i(Y_i) - e_i(\mu_i(X_i))\right] \cdot \frac{\xi_i}{\pi_i(X_i)},$$

$$E_n^{\text{ppi}} := \prod_{i=1}^n e_i^{\text{ppi}}, \qquad (\xi_i \sim \text{Bern}(\pi_i(X_i))).$$

This construction is motivated by the fact that, conditional on all data prior to the time point $i$, the prediction-powered e-value components $e_i^{\text{ppi}}$ match the non-prediction-powered ones $e_i$ in expectation:

$$\mathbb{E}_i\left[e_i(\mu_i(X_i)) + \left[e_i(Y_i) - e_i(\mu_i(X_i))\right] \cdot \frac{\xi_i}{\pi_i(X_i)}\right]$$

$$= \mathbb{E}_i[e_i(\mu_i(X_i))] + \mathbb{E}_i\left[\left[e_i(Y_i) - e_i(\mu_i(X_i))\right] \cdot \frac{\xi_i}{\pi_i(X_i)}\right]$$

$$= \mathbb{E}_i[e_i(\mu_i(X_i))]$$

$$+ \mathbb{E}_i\left[\left[e_i(Y_i) - e_i(\mu_i(X_i))\right] \cdot \frac{\xi_i}{\pi_i(X_i)} \mid \xi_i = 1\right] \mathbb{P}_i[\xi_i = 1]$$

$$+ \mathbb{E}_i\left[\left[e_i(Y_i) - e_i(\mu_i(X_i))\right] \cdot \frac{\xi_i}{\pi_i(X_i)} \mid \xi_i = 0\right] \mathbb{P}_i[\xi_i = 0]$$

$$= \mathbb{E}_i[e_i(\mu_i(X_i))] + \mathbb{E}_i[e_i(Y_i) - e_i(\mu_i(X_i))] = \mathbb{E}_i[e_i(Y_i)].$$

Furthermore, the boundedness of the e-values' components and on the $\pi$ ensure that the quantity is always nonnegative. Using these facts along with a backward induction argument, one can prove:

**Theorem 2.1.** $E_n^{\text{ppi}}$ *is a valid e-value for the null $H_0$. Additionally:*

*(i) If $(E_0, E_1, \ldots)$ form a test supermartingale – i.e., a nonnegative supermartingale with $\mathbb{E}[E_0] \leq 1$ under the null $H_0$ – then so is $(E_0^{\text{ppi}}, E_1^{\text{ppi}}, \ldots)$;*

*(ii) More generally, if $(E_0, E_1, \ldots)$ form an e-process – i.e., a nonnegative stochastic process such that for all stopping times $\tau$, the null $H_0$ implies that $\mathbb{E}[E_\tau] \leq 1$ – then so is $(E_0^{\text{ppi}}, E_1^{\text{ppi}}, \ldots)$ for all finite stopping times.*

Besides having valid e-values – which assures us of type-I error control – one should check whether they are efficient/powerful. We can check that, under mild assumptions, our e-process has good power in terms of the expected growth rate (Kelly, 1956) as long as the models $\mu_i$ match the true data $Y_i$ sufficiently well:

**Theorem 2.2.** *Suppose that the $e_i(\cdot)$ are each $L_i$-Lipschitz, and that $\pi_i(X_i) \geq 1 - a_i/b_i + \epsilon_i$ for some $\epsilon_i > 0$, for all $i$. Then there exists some constant $c > 0$ independent of $n$ such that*

$$\mathbb{E}\left[\frac{1}{n} \log E_n^{\text{ppi}}\right] \geq \mathbb{E}\left[\frac{1}{n} \log E_n\right] - \frac{c}{n} \sum_{i=1}^n \mathbb{E}[\|\mu_i(X_i) - Y_i\|].$$

More general and precise statements are also possible, but less compact; see Theorems A.6 in the appendix.

The sequential nature of the prediction-powered e-values – which holds regardless of whether the original e-values were of sequential nature – allows for an extremely versatile procedure. For instance, in contrast to most existing prediction-powered inference procedures, we are able to update both our underlying prediction model and our data collection rule over the course of our inference process, with no restrictions other than not using future information and having to satisfy the boundedness assumptions.

The resulting algorithm for hypothesis testing is remarkably simple to implement, given its generality. The pseudocode can be found in Algorithm 1.

**Algorithm 1** Prediction-Powered E-Values

>   **Input:** base e-value components $(e_1(\cdot), e_2(\cdot), \dots)$
>   **Output:** prediction-powered e-values $(E_0^{\text{ppi}}, E_1^{\text{ppi}}, \dots)$
>   $E_0^{\text{ppi}} \leftarrow 1$
>   Initialize $\mu : \mathcal{X} \to \mathcal{Y}$ and $\pi : \mathcal{X} \to [1 - a_1/b_1, 1]$
>   **for** each $i = 1, 2, \dots$ **do**
>     Get 'cheap' data $X_i$
>     Sample $\xi_i \sim \text{Bern}(\pi(X_i))$
>     **if** $\xi_i = 1$ **then**
>       Collect 'expensive' data $Y_i$
>       $E_i^{\text{ppi}} \leftarrow E_{i-1}^{\text{ppi}} \cdot \frac{e_i(Y_i) - (1 - \pi(X_i))e_i(\mu(X_i))}{\pi(X_i)}$
>     **else**
>       $E_i^{\text{ppi}} \leftarrow E_{i-1}^{\text{ppi}} \cdot e_i(\mu(X_i))$
>     **end if**
>     Optionally update $\pi$ and $\mu$
>   **end for**

### 2.2. From hypothesis testing to confidence intervals

With prediction-powered e-values in hand, we can easily produce prediction-powered confidence intervals/sequences by considering a family of e-values indexed by the parameter in question.

Suppose we want to produce a confidence interval/sequence for a parameter $\theta^\star \in \Theta$ of the data generating process, and consider the family of nulls $H_0^{(\theta)} : \theta^\star = \theta$, indexed by $\theta$. For each such null, we can construct a corresponding prediction-powered e-value $E_n^{\text{ppi}-(\theta)}$ and then consider the set

$$C_n^{\text{ppi}-(\alpha)} := \left\{ \theta \in \Theta : E^{\text{ppi}-(\theta)} < 1/\alpha \right\}.$$

By the standard duality between hypothesis tests and confidence sets, it then holds that:

**Proposition 2.3.** $C_n^{\text{ppi}-(\alpha)}$ *is a valid confidence interval – i.e.,* $\mathbb{P}[\theta^\star \in C_n^{\text{ppi}-(\alpha)}] \geq 1 - \alpha$. *Moreover:*

*(i) If the underlying e-values form a nonnegative super-martingale, then the prediction-powered intervals are anytime-valid (also known as confidence sequences):* $\mathbb{P}[\forall n \in \mathbb{N}, \ \theta^\star \in C_n^{\text{ppi}-(\alpha)}] \geq 1 - \alpha$*;*

*(ii) More generally, if the underlying e-values form e-processes, then the prediction-powered intervals are valid at arbitrary stopping times:* $\mathbb{P}[\theta^\star \in C_\tau^{\text{ppi}-(\alpha)}] \geq 1 - \alpha$ *for any stopping time* $\tau$*.*

Again, we are also interested in how efficient these confidence sequences are. Just like before, as long as our predictive models are good, we get more concentrated intervals, as measured by the area under the log-p-landscape:

**Proposition 2.4.** *Under the assumptions of Theorem 2.2, let* $\nu$ *be a measure over the parameter space* $\Theta$*. Then there*

*exists some c for which*

$$\mathbb{E}\left[\int \frac{1}{n} \log \frac{1}{E_n^{\text{ppi}-(\theta)}} \mathrm{d}\nu(\theta)\right] \leq \mathbb{E}\left[\int \frac{1}{n} \log \frac{1}{E_n^{(\theta)}} \mathrm{d}\nu(\theta)\right]$$
$$+ \frac{\nu(\Theta)c}{n} \sum_{i=1}^n \mathbb{E}[\|\mu_i(X_i) - Y_i\|].$$

These results may be mapped to the actual measure of the confidence interval, but this is nontrivial; see the Appendix.

### 2.3. General e-value-powered algorithms

Beyond simple hypothesis testing and confidence sequences, e-values can also be used as components of more elaborate inference procedures, for example in causal discovery (e.g. (Peters et al., 2017)), change point detection (Shin et al., 2022; Shekhar & Ramdas, 2023a;b) and test-time adaptation (Bar et al., 2024). Our prediction-powered e-values can also be seamlessly integrated into such procedures.

Consider that we have a family of e-values $E_n^{(\gamma)}$ for respective nulls $H_0^{(\gamma)}$, indexed by $\gamma \in \Gamma$, and our overall algorithm is of the form $\mathcal{A}((E_n^{(\gamma)})_{\gamma \in \Gamma})$. Moreover, our algorithm comes endowed with some 'validity' property, and is such that this validity depends only on the inputted e-values being valid:

**Assumption 2.5.** *If* $E_n^{(\gamma)}$ *is a valid e-value for the null* $H_0^{(\gamma)}$ *for every* $\gamma \in \Gamma$*, then* $\mathcal{A}((E_n^{(\gamma)})_{\gamma \in \Gamma})$ *is valid.*

It is then easy to show that by simply replacing the input e-values by their prediction-powered counterparts, the validity property is maintained:

**Proposition 2.6.** *Under Assumption 2.5, it holds that* $\mathcal{A}((E_n^{\text{ppi}-(\gamma)})_{\gamma \in \Gamma})$ *is also* valid*. If the underlying e-values are e-processes, then it further holds that* $\mathcal{A}((E_\tau^{\text{ppi}-(\gamma)})_{\gamma \in \Gamma})$ *is* valid *for any finite stopping time* $\tau$*.*

It is still also of interest to consider some notion of 'power' or 'efficiency' of the resulting prediction-powered procedure. However, such an analysis needs to consider more of the particular algorithm in principle, and so should be done on a case-by-case basis. Similarly, the appropriate notion of anytime-validity (which would be implied by the underlying e-values forming test supermartingales) depends on the particular definition of validity for the algorithm in question and so should be considered in a case-by-case basis. Nevertheless, the case of an e-process still holds generally.

### 2.4. The Asymptotic Case

Though typically considered in the context of nonasymptotic statistics, e-values also have asymptotic analogues (Waudby-Smith et al., 2021; Ramdas & Wang, 2024). We focus

on the main text only on nonasymptotic e-values, but our ideas directly map to the asymptotic setting just as well; see Appendix B.1.

# 3. Experiments and Case Studies

In this section we present four case studies where we use our method, highlighting the modifications made to the base methods in the process of prediction-empowerment.

## 3.1. Estimation of a Mean: Prevalence of Diabetes from Survey Data

In this first case study we seek to estimate the prevalence of diabetes on a cohort, upon which we work atop the dataset of (CDC, 2015). This estimation is key to the scaling of resources in health systems, as this medical condition can be very common and very costly to treat in many populations.

Actually assessing the presence of diabetes can be somewhat costly, requiring thorough analysis of individual medical records. On the other hand, we have readily available data in the form of short survey responses, consisting of simple questions such as "do you have high blood pressure?", "do you have high cholesterol?", "have you smoked at least 100 cigarettes in your entire life?", and so on (see Appendix C for the full list). Considering that these questions capture health indicators that are fairly predictive of diabetes, it is appealing to leverage them in a prediction-powered manner.

More formally, we have a data stream $(X_i, Y_i)_{i=1}^{\infty}$ where the $X_i$ correspond to the responses to our survey questions, and the $Y_i$ correspond to a binary indicator of whether the individual is diabetic. For the sake of evaluation, our dataset includes all $Y_i$, but in a real-world setting it would be expected that they would be largely missing; we will simulate this missingness. Our goal is to infer the mean

$$\text{prevalence of diabetes} = \mathbb{E}[\mathbb{1}[Y_i = \text{diabetic}]].$$

This is the mean of a random variable bounded in $[0, 1]$, and so we can use the e-value-based method for inference of bounded means of (Waudby-Smith & Ramdas, 2020). Our confidence interval/sequence is thus given by the set

$$C_n^{(\alpha)} = \left\{ \theta \in [0, 1] : E_n^{(\theta)} < 1/\alpha \right\},$$

for

$$E_n^{(\theta)} = \prod_{i=1}^{n} \left( 1 + \lambda_i \left( \mathbb{1}[Y_i = \text{diabetic}] - \theta \right) \right),$$

where $(\lambda_i)_{i=1}^{\infty}$ is a predictable sequence of bets bounded in $(-\frac{1}{1-\theta}, \frac{1}{\theta})$. In particular, each $E_n^{(\theta)}$ is a test supermartingale – and thus a sequence of e-values – for a corresponding null $H_0^{(\theta)}$ : prevalence of diabetes $= \theta$ (Waudby-Smith & Ramdas, 2020).

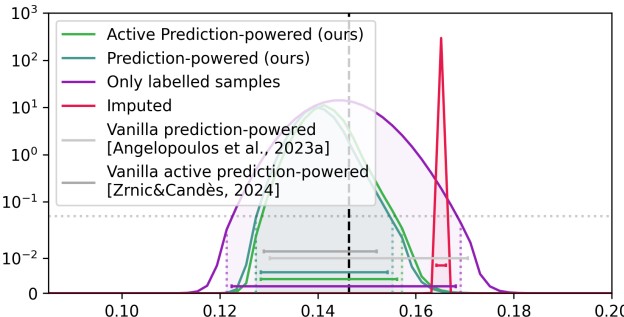

*Figure 1.* **Prediction-powered confidence sequences.** The plot shows the p-landscape (i.e., parameter on the x-axis, reciprocal of the e-value on the y-axis) for the confidence sequence generated by our method (green), along with those for inference using only labelled samples (purple) and by using an imputation approach. The 95% confidence intervals for each p-landscape (i.e., region where the p-landscape is above 0.05) is shaded. Our method provides the tightest valid nonasymptotic intervals, comparable to the active asymptotic method of (Zrnic & Candes, 2024); using only the labelled samples or vanilla PPI (Angelopoulos et al., 2023a) yields weaker inferences, and using imputation fails to cover the true mean.

These e-values are already in our required form of Equation (1), but additional care needs to be taken with regards to the bounds of the e-values' components. As-is, the components are bounded just in $[0, 1 + \max\{\theta/(1-\theta), (1-\theta)/\theta\}]$. This means that we would require the data collection probabilities $\pi_i(X_i)$ to be bounded in $[1, 1]$ – i.e., we would always need to collect data; this is clearly insufficient for our purposes.

Fortunately, we have a direct way of controlling these bounds by the means of the bets ($\lambda_i$). If, instead of requiring them to be bounded in $(-\frac{1}{1-\theta}, \frac{1}{\theta})$, we require them to be bounded in $(-\frac{c}{1-\theta}, \frac{c}{\theta})$ for some $0 < c \leq 1$, then we have that the components are bounded in $[1 - c, 1 + c\max\{\theta/(1-\theta), (1-\theta)/\theta\}]$, which now leads to nontrivial bounds on the $\pi_i(X_i)$. In particular, for any desired lower bound $\pi_{\inf}$ for $\pi_i(X_i)$, we can now solve for some $c$ for which

$$1 - \frac{1 - c}{1 + c\max\{\theta/(1-\theta), (1-\theta)/\theta\}} \leq \pi_{\inf}, \quad (2)$$

satisfying our requirements; we use $\pi_{\inf} = 1\%$.

We then have the following methods for doing inference with a fixed labelling budget $\pi_{\inf}$:

- **Only labelled samples:** collect $\lfloor \pi_{\inf} \cdot n \rceil$ labelled samples, and use the standard, non-prediction-powered e-values of (Waudby-Smith & Ramdas, 2020) to estimate the mean. For the bets $\lambda_i$, we use the aGRAPA method proposed by (Waudby-Smith & Ramdas, 2020), bounded to $(-\frac{1}{1-\theta}, \frac{1}{\theta})$;

- **Prediction-powered (ours):** use our prediction-powered e-values method atop the e-value with bets truncated as per Equation (2). The predictive model is updated over the course of the inference, whenever we get a new data label. For the collection probabilities $\pi_i$, we always yield $\pi_{\inf}$, the lowest possible value, in an effort to minimize data collection costs. The predictive model is updated over the course of the procedure by retraining on the augmented dataset

- **Active prediction-powered (ours):** same as the previous 'prediction-powered' method, but with a different choice of collection probabilities $\pi_i$. This time, rather than opting for constant, always as low as possible, probabilities, we follow an approximately optimal choice which takes into consideration the $X_i$, as delineated in Appendix B.2. This gives an 'active inference'/'active learning' flavor to our method.

- **Imputation:** we simply learn a predictive model to predict the missing $Y_i$ from the available $X_i$, and impute the missing $Y_i$ with it without any care to use some prediction-powered inference method. This will often yield invalid inferences, but is very common in practice and thus a relevant baseline.

- **Vanilla prediction-powered:** for the sake of comparison to prior work, we also consider the method of (Angelopoulos et al., 2023a). This method requires the prediction model to be fixed a priori, so we first split the collected labels in a training set to train it, and use the remaining labels for their prediction-powered inference method. For confidence intervals, we use CLT-based ones as proposed by the authors.

- **Vanilla active prediction-powered:** we also compare to the method of (Zrnic & Candes, 2024), which does prediction-powered inference in an active setting, with a similar AIPW approach. We use CLT-based CIs for their estimator, as they propose, and update the underlying predictive model as we do for our method.

Figure 1 shows the result of our experiment. Our prediction-powered methods provides valid confidence intervals that are tighter and more concentrated around the true mean in comparison to only using labelled samples, while the imputation approach is strongly concentrated away from the true mean, and would lead to invalid conclusions. In comparison to the method of (Angelopoulos et al., 2023a), our method

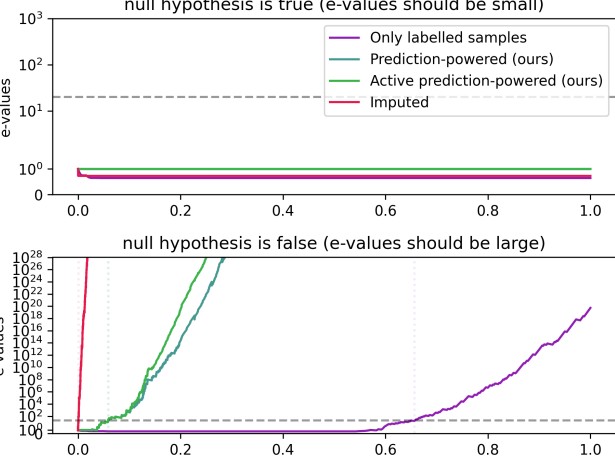

*Figure 2.* **Prediction-powered anytime-valid hypothesis testing.** The plot shows the e-values over time for testing two null hypotheses – one on the bottom, which should be rejected, and one on top, which should not be rejected. Our prediction-powered e-values provide the strongest valid signal for rejection ($E \geq 20$ for a significance level of 95%, marked by the dashed lines), as the imputation approach rejects before the null is actually violated; for non-rejection ($E < 20$), all the methods appear valid, but ours still attains the highest e-value.

provides tighter intervals in spite of its nonasymptotic nature, likely due to its ability to train the predictive model without a data split; indeed, when compared to the baseline of (Zrnic & Candes, 2024), which also updates the models, we see a similar performance obtained by our method, but now with the stronger guarantees of e-values.

### 3.2. Testing the Online Risk: Online Monitoring of a Deployed Model for Forest Cover Prediction

For our second experiment, we consider the task of monitoring the risk of a predictive model for forest cover types online. Forest cover prediction is of wide use in remote sensing tasks and particularly for tracking of deforestation and land use, which is, in turn, very useful for climate research. Moreover, online risk monitoring is ubiquitous and applicable to any setting where a predictive model is involved.

Again we have a data stream $(X_i, Y_i)_{i=1}^{\infty}$, where $X_i$ indicate input variables to our predictive model – in this case, corresponding to data from satellite images – and $Y_i$ are the labels denoting the corresponding cover type (which is a categorical variable). Naturally, $Y_i$ is generally missing – after all, if it weren't, then we would have no need to predict it. In our experiment, we work on the dataset of (Blackard, 1998). For the sake of evaluation, we have access to all $Y_i$, but will simulate the missingness. The notion of risk in which we are interested is given by the 0-1 loss: $\text{Risk}_i(f) = \mathbb{E}[\mathbb{1}[f(X_i) \neq Y_i]]$. We have already trained

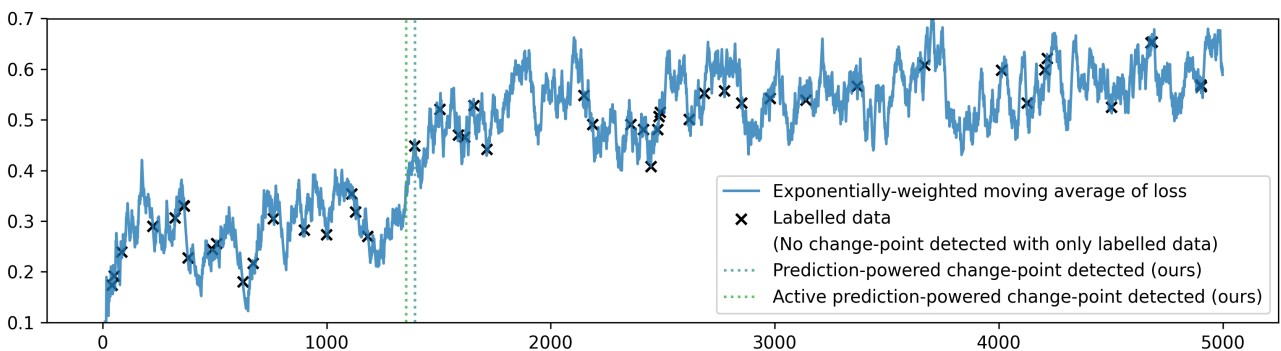

*Figure 3.* **Prediction-powered change-point detection via e-values.** The plot shows the exponential moving average of a time series (in blue), with the few collected labels denoted by the scattered Xs. Our prediction-powered methods detect the change-point accurately, while the base method that only considers the labelled data points does not detect any change-point.

the predictive model $f$ independent of our data stream (in the case of our experiment, in a separate training split) and have similarly evaluated it on a separate validation set, also independent of our data stream, upon which we obtained a validation 0-1 loss of ValRisk. For continuous risk monitoring, we want to test the null hypothesis that

$$H_0 : \text{Risk}_i(f) \leq \text{ValRisk} + \epsilon_{\text{tol}}, \quad \text{for all } i = 1, 2, \ldots,$$

for some tolerance level $\epsilon_{\text{tol}}$, for example equal to $0.05$. In particular, we would like for this hypothesis test to be anytime-valid, so that at any point we can reach safe conclusions from it.

Inspired by the work of (Podkopaev & Ramdas, 2021), we consider the following e-value:

$$E_n := \prod_{i=1}^{n} \left( 1 + \lambda_i \Big( \mathbb{1}[f(X_i) \neq Y_i] - (\text{ValRisk} + \epsilon_{\text{tol}}) \Big) \right),$$
(3)

where $(\lambda_i)_{i=1}^{\infty}$ is a predictable sequence of bets bounded in $\big[0, 1/(\text{ValRisk} + \epsilon_{\text{tol}})\big)$. This forms a test supermartingale for the null $H_0$.

Much like in the example of inference of the prevalence of diabetes in Section 3.1, the e-values are already of the desired form, but additional care must be taken with regard to the limits of the components. As-is, they are bounded in $[0, 1 + \max\{1/(\text{ValRisk} + \epsilon_{\text{tol}}) - 1, 0\}]$, meaning that our collection probabilities would have to be within $[1, 1]$. Similarly to what we did in Section 3.1, we tweak the bounds for the bets $\lambda_i$ to make them bounded within $\big[0, c/(\text{ValRisk} + \epsilon_{\text{tol}})\big)$ for some $0 < c \leq 1$, leading to components bounded in $[1 - c, 1 + c \max\{1/(\text{ValRisk} + \epsilon_{\text{tol}}) - 1, 0\}]$. We can then solve for the $c$ that satisfies

$$1 - \frac{1 - c}{1 + c \max\{1/(\text{ValRisk} + \epsilon_{\text{tol}}) - 1, 0\}} \leq \pi_{\text{inf}}, \quad (4)$$

for a desired labelling budget $\pi_{\text{inf}}$, which we take to be equal $0.5\%$.

The methods we consider for our experiment are akin to those of Section 3.1:

- **Only labelled samples** at every data point $i$, we sample $\xi_i \sim \text{Bern}(\pi_{\text{inf}})$. If $\xi_i = 1$, then we collect that data point and update the non-prediction-powered e-value in Equation (3). Since the data collection is sampled independently of all else, this is a valid e-value, and forms a test supermartingale; moreover, only about $\approx \pi_{\text{inf}} \cdot n$ samples will be collected. However, only data points where $\xi_i = 1$ are used for inference.

- **Prediction-powered (ours):** we compute the prediction-powered e-value atop the base-evalue in Equation (3) tweaked to satisfy the boundedness conditions as per Equation (4). We then have two predictive models: one which is the predictive model whose risk we want to monitor $- \mu -$ and another which is used for prediction-powered inference, which receives $X_i$ and predicts the 0-1 loss for that point, $\mathbb{1}[\mu(X_i) \neq Y_i]$. The first model $\mu$ is held static over the course of the inference, while the one for prediction-powered inference is updated whenever we collect a new label. Collection probabilities $\pi_i(X_i)$ are held constant at $\pi_{\text{inf}}$, leading to label collection matching the baseline of only using labelled samples.

- **Active prediction-powered (ours):** the same as our 'non-active' prediction-powered method, but label collection probabilities are given by the approximately optimal choice presented in Appendix B.2.

- **Imputation:** as a final baseline, we consider simply imputing the 0-1 loss at points where we have not

collected the true label, with no regard to prediction-powreed inference. This is invalid in general, but commonly used in practice.

Note that standard prediction-powered inference methods (e.g., from (Angelopoulos et al., 2023a)) are no longer directly applicable due to the requirement of anytime-validity, as well as the fact that our hypothesis test does not come from a two-sided test for a mean (which would then be an instance of simple Z-estimation).

To fully assess the hypothesis test, we consider two settings here. In the first setting, there is no change in distribution: the data stream for the inference follows the exact same distribution as training and validation, and thus the null hypothesis should hold. For the second setting, we increasingly poison the labels over the course of time to simulate distribution drift.

The results can be seen in Figure 2. Without data poisoning, none of the methods reject the hypothesis, which is appropriate; though it is interesting to note that our prediction-powered methods were the ones with the highest e-values, managing to stay at around 1. Under data poisoning, both of our prediction-powered methods detect the distribution drift much quicker than the method that only uses labelled samples, despite both having access to the same labelled samples. The active prediction-powered method seems to reject the hypothesis a tiny bit earlier and yields larger e-values (i.e., with more evidence towards rejection), at the cost of just a tiny bit more data. The imputation method seemingly detects the shift even earlier, but does so before the null hypothesis is actually falsified; thus, it produces a false alarm with extremely high confidence.

### 3.3. Change-Point Detection: Detecting Changes in the Quality of a Deployed Model

Still in the context of testing the cover prediction model of Section 3.2, we now consider not just detecting when the risk goes below a certain level, but detecting *any* change. E-values have seen good use in the change-point detection literature (Shin et al., 2022; Shekhar & Ramdas, 2023a;b); we opt here for the method proposed in (Shekhar & Ramdas, 2023a), where change-point detection is reduced to a simple algorithm atop confidence sequences initialized at each time step. For the underlying confidence sequences, we use the same ones as in Section 3.1. Compared to Section 3.2, the only change we make to the data is the introduction of a crisp change-point for better visualization.

Figure 3 displays a high-frequency exponentially moving average of our data (to give a notion of the underlying data stream) that uses data at all points, regardless of whether they are accessible to the analyst; scattered throughout are the few data points that were labelled and that the analyst

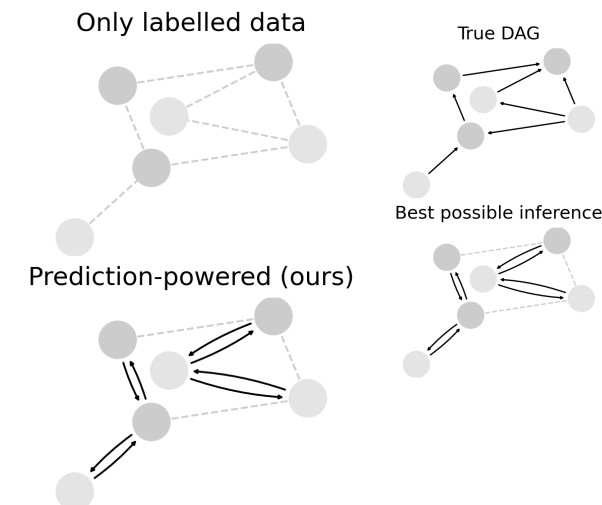

*Figure 4.* **Prediction-powered causal discovery with e-values.** We compare our prediction-powered causal discovery method with one that uses only labelled data. The lighter nodes correspond to the costly variables, while the darker nodes correspond to cheaper readily-available ones. The standard base method does not detect any edges in the causal graph (denoted by the dashed edges), while ours detects as many edges as the 'best possible' method, which uses all the data points regardless of data acquisition costs.

does have access to. Our prediction-powered method detects the change-point accurately while retaining the strong guarantees of (Shekhar & Ramdas, 2023a), whereas the non-prediction-powered baseline that uses only available labelled data fails to detect any change-point.

### 3.4. Causal Discovery: Constraint-Based Structure Learning with Costly Covariates

Causal inference is of essence to any area where one plans interventions, but the usual methods require knowledge of a DAG describing (a simplified version of) the data generating process. Causal discovery (a.k.a. causal structure learning) methods seek to learn this from data. Some particularly common methods for causal learning include the PC (Burr, 2003) and FCI (Spirtes et al., 1995) algorithms; all of these belong to the class of so called 'constraint-based' structure learning, where the DAG is inferred by the means of many hypothesis tests for conditional independencies. In spite of potential multiple comparison concerns, these algorithms are generally said to be valid as long as the underlying hypothesis tests are valid (i.e., control type-I error).

In this section we consider the problem of causal discovery with the PC algorithm (Burr, 2003) with some costly covariates, which will be generally missing. As is usual in the causal discovery literature, we evaluate on synthetic

data generated with a randomly generated DAG, in order to have access to the true DAG. Our DAG features 6 variables, of which 3 are considered costly. Overall, our cheap data $X_i$ consists of the 3 always-available variables, whereas the costly data $Y_i$ consists of the 6 full variables. For constraint-based causal learning, we need to be able to test hypotheses of the form

$$H_0^{(A,B,C)} : A \perp\!\!\!\perp B \mid C,$$

where $A$, $B$ and $C$ consist of subsets of our 6 variables, possibly empty.

There do exist sequential e-value tests for conditional independence (Shaer et al., 2022; Grünwald et al., 2022), but they work under the Model-X framework, which requires knowledge of conditionals that are typically inaccessible in the context of causal discovery. We thus opt instead for Fisher's z-transformation of partial correlation test, which is commonly used in causal discovery implementations (e.g., (Markus Kalisch et al., 2012; Zheng et al., 2024)). But it is based on p-values, is not of sequential nature, is asymptotic, and works atop rather heavy normality assumptions.

We first need to adapt it to our required form, following Equation (1). To do so, we first rearrange our data stream $(X_i, Y_i)_{i=1}^{\infty}$ to arrive in batches of $B$ samples, $(X_j^{\text{batch}}, Y_j^{\text{batch}})_{j=1}^{\infty}$; these batches will be the unit of data for our prediction-powered procedure. We can then compute the test's p-value for each batch, and calibrate this p-value into an e-value by the means of the following PToE calibrator (Vovk & Wang, 2019):

$$\text{PToE}(p) = \frac{1 - p + p \log p}{p(-\log p)^2}.$$

To ensure that our e-values' components are appropriately bounded, we first clip the p-values (prior to calibration) to lie within $(10^{-7}, 1]$ (so that they are bounded at all; this clipping preserves the validity of the p-values), and then rescale the calibrated e-values by the means of a rescaling function

$$\text{rescale}_\eta(e) := \eta \cdot (e - 1) + 1,$$

with $\eta$ chosen so as to satisfy a labelling budget of $\pi_{\text{inf}} = 10\%$ (as in the previous sections). Because the p-values are only valid asymptotically, the batch size $B$ cannot be too small; we use $B = 100$.

The results can be seen in Figure 4. When using only labelled data according to our data collection budget, the causal discovery method identifies no edges at all. By using our prediction-powered e-values, we detect over half of the edges, matching the best possible scenario (i.e., what would happen if we had access to the whole dataset). In terms of the average structural Hamming distance over various sampled graphs, using only labelled data we obtain an average 12.5; our method halves this to 6.7, and the best possible scenario would obtain that of 6.4.

## Acknowledgements

This research is funded by Canada's International Development Research Centre (IDRC) (Grant No. 109981) and UK International Development. CJS acknowledges financial support from CNPq and FAPERJ.

## Impact Statement

This paper presents work whose goal is to advance the field of Machine Learning and Statistics. There are many potential societal consequences of our work, none which we feel must be specifically highlighted here.

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

# A. Proofs

Throughout, we denote by $\mathcal{F}_i$ the $i$-th element of the underlying data filtration.

**Theorem A.1** (Theorem 2.1 in the main text). $E_n^{\mathrm{ppi}}$ *is a valid e-value for the null* $H_0$.
*Additionally:*

(i) *If* $(E_0, E_1, \ldots)$ *form a test supermartingale – i.e., a nonnegative supermartingale with* $\mathbb{E}[E_0] \leq 1$ *under the null* $H_0$ – *then so is* $(E_0^{\mathrm{ppi}}, E_1^{\mathrm{ppi}}, \ldots)$;

(ii) *More generally, if* $(E_0, E_1, \ldots)$ *form an e-process – i.e., a nonnegative stochastic process such that for all stopping times* $\tau$, *the null* $H_0$ *implies that* $\mathbb{E}[E_\tau] \leq 1$ – *then so is* $(E_0^{\mathrm{ppi}}, E_1^{\mathrm{ppi}}, \ldots)$ *for all finite stopping times.*

*Proof.* First, note that $E_n^{\mathrm{ppi}}$ is always nonnegative for all $n \in \mathbb{N}$: by induction, it holds for $n = 0$ (where $E_n^{\mathrm{ppi}} = E_0^{\mathrm{ppi}} = 1$), and, for the inductive step,

$$E_{n+1}^{\mathrm{ppi}} = E_n^{\mathrm{ppi}} \cdot \left( e_{n+1}(\mu_{n+1}(X_{n+1})) + \left[ e_{n+1}(Y_{n+1}) - e_{n+1}(\mu_{n+1}(X_{n+1})) \right] \cdot \frac{\xi_{n+1}}{\pi_{n+1}(X_{n+1})} \right) \geq 0$$

$$\iff e_{n+1}(\mu_{n+1}(X_{n+1})) + \left[ e_{n+1}(Y_{n+1}) - e_{n+1}(\mu_{n+1}(X_{n+1})) \right] \cdot \frac{\xi_{n+1}}{\pi_{n+1}(X_{n+1})} \geq 0;$$

If $\xi_{n+1} = 0$, then the left-hand-side equals $e_{n+1}(\mu_{n+1}(X_{n+1})) \geq a_{n+1} > 0$. Otherwise, it equals

$$\frac{e_{n+1}(Y_{n+1}) - (1 - \pi_{n+1}(X_{n+1}))e_{n+1}(\mu_{n+1}(X_{n+1}))}{\pi_{n+1}(X_{n+1})} \geq \frac{a_{n+1} - (1 - \pi_{n+1}(X_{n+1}))b_{n+1}}{\pi_{n+1}(X_{n+1})} \geq 0$$

$$\iff a_{n+1} - (1 - \pi_{n+1}(X_{n+1}))b_{n+1} \geq 0 \iff a_{n+1} \geq (1 - \pi_{n+1}(X_{n+1}))b_{n+1}$$

$$\iff 1 - a_{n+1}/b_{n+1} \leq \pi_{n+1}(X_{n+1}),$$

which holds by construction.

So all that remains is to show that its properties under the null hold. Hence, from here on out, we assume that the null $H_0$ is true.

We will first show that, for any $n \in \mathbb{N}$, $\mathbb{E}[E_n^{\mathrm{ppi}}] \leq 1$. To do so, we will first prove the following lemma by backward induction:

**Lemma A.2.** *Let* $n \in \mathbb{N}$ *and $A$ denote an event. Then, for any* $1 \leq k \leq n$, *it holds that* $\mathbb{E}[\prod_{i=k}^{n} e_i^{\mathrm{ppi}} \mid A, \mathcal{F}_{k-1}] = \mathbb{E}[\prod_{i=k}^{n} e_i(Y_i) \mid A, \mathcal{F}_{k-1}]$

*Proof.* The base case is when $k = n$. Then

$$\mathbb{E}\left[ \prod_{i=k}^{n} e_i^{\mathrm{ppi}} \mid A, \mathcal{F}_{k-1} \right]$$

$$= \mathbb{E}\left[ e_n^{\mathrm{ppi}} \mid A, \mathcal{F}_{n-1} \right] = \mathbb{E}\left[ e_n(\mu_n(X_n)) + \frac{\xi_n}{\pi_n(X_n)} \left( e_n(Y_n) - e_n(\mu_n(X_n)) \right) \mid A, \mathcal{F}_{n-1} \right]$$

$$= \mathbb{E}\left[ e_n(\mu_n(X_n)) \mid A, \mathcal{F}_{n-1} \right] + \mathbb{E}\left[ \frac{\xi_n}{\pi_n(X_n)} \left( e_n(Y_n) - e_n(\mu_n(X_n)) \right) \mid \xi_n = 1, A, \mathcal{F}_{n-1} \right] \mathbb{P}[\xi_n = 1 \mid A, \mathcal{F}_{n-1}]$$

$$+ \mathbb{E}\left[ \frac{\xi_n}{\pi_n(X_n)} \left( e_n(Y_n) - e_n(\mu_n(X_n)) \right) \mid \xi_n = 0, A, \mathcal{F}_{n-1} \right] \mathbb{P}[\xi_n = 0 \mid A, \mathcal{F}_{n-1}]$$

$$= \mathbb{E}\left[ e_n(\mu_n(X_n)) \mid A, \mathcal{F}_{n-1} \right] + \mathbb{E}\left[ \frac{1}{\pi_n(X_n)} \left( e_n(Y_n) - e_n(\mu_n(X_n)) \right) \mid A, \mathcal{F}_{n-1} \right] \pi_n(X_n)$$

$$= \mathbb{E}\left[ e_n(\mu_n(X_n)) \mid A, \mathcal{F}_{n-1} \right] + \mathbb{E}\left[ e_n(Y_n) - e_n(\mu_n(X_n)) \mid A, \mathcal{F}_{n-1} \right]$$

$$= \mathbb{E}\left[ e_n(Y_n) \mid A, \mathcal{F}_{n-1} \right] = \mathbb{E}\left[ \prod_{i=k}^{n} e_i(Y_i) \mid A, \mathcal{F}_{k-1} \right].$$

For the induction step, given that the hypothesis holds for $k + 1 \leq n$, we want to show that it holds for $k$. It follows, using the law of total expectation:

$$\mathbb{E}\left[\prod_{i=k}^{n} e_i^{\mathrm{ppi}} \mid A, \mathcal{F}_{k-1}\right]$$

$$= \mathbb{E}\left[e_k^{\mathrm{ppi}} \prod_{i=k+1}^{n} e_i^{\mathrm{ppi}} \mid A, \mathcal{F}_{k-1}\right] = \mathbb{E}\left[\mathbb{E}\left[e_k^{\mathrm{ppi}} \prod_{i=k+1}^{n} e_i^{\mathrm{ppi}} \mid A, \mathcal{F}_k\right] \mid A, \mathcal{F}_{k-1}\right]$$

$$= \mathbb{E}\left[e_k^{\mathrm{ppi}} \mathbb{E}\left[\prod_{i=k+1}^{n} e_i^{\mathrm{ppi}} \mid A, \mathcal{F}_k\right] \mid A, \mathcal{F}_{k-1}\right] = \mathbb{E}\left[e_k^{\mathrm{ppi}} \mathbb{E}\left[\prod_{i=k+1}^{n} e_i(Y_i) \mid A, \mathcal{F}_k\right] \mid A, \mathcal{F}_{k-1}\right]$$

$$= \mathbb{E}\left[\left(e_k(\mu_k(X_k)) + \frac{\xi_k}{\pi_k(X_k)}\left(e_k(Y_k) - e_k(\mu_k(X_k))\right)\right) \mathbb{E}\left[\prod_{i=k+1}^{n} e_i(Y_i) \mid A, \mathcal{F}_k\right] \mid A, \mathcal{F}_{k-1}\right]$$

$$= \mathbb{E}\left[e_k(\mu_k(X_k)) \mathbb{E}\left[\prod_{i=k+1}^{n} e_i(Y_i) \mid A, \mathcal{F}_k\right] \mid A, \mathcal{F}_{k-1}\right]$$

$$\quad + \mathbb{E}\left[\frac{\xi_k}{\pi_k(X_k)}\left(e_k(Y_k) - e_k(\mu_k(X_k))\right) \mathbb{E}\left[\prod_{i=k+1}^{n} e_i(Y_i) \mid A, \mathcal{F}_k\right] \mid \xi_k = 1, A, \mathcal{F}_{k-1}\right] \mathbb{P}[\xi_k = 1 \mid A, \mathcal{F}_{k-1}]$$

$$\quad + \mathbb{E}\left[\frac{\xi_k}{\pi_k(X_k)}\left(e_k(Y_k) - e_k(\mu_k(X_k))\right) \mathbb{E}\left[\prod_{i=k+1}^{n} e_i(Y_i) \mid A, \mathcal{F}_k\right] \mid \xi_k = 0, A, \mathcal{F}_{k-1}\right] \mathbb{P}[\xi_k = 0 \mid A, \mathcal{F}_{k-1}]$$

$$= \mathbb{E}\left[e_k(\mu_k(X_k)) \mathbb{E}\left[\prod_{i=k+1}^{n} e_i(Y_i) \mid A, \mathcal{F}_k\right] \mid A, \mathcal{F}_{k-1}\right]$$

$$\quad + \mathbb{E}\left[\frac{1}{\pi_k(X_k)}\left(e_k(Y_k) - e_k(\mu_k(X_k))\right) \mathbb{E}\left[\prod_{i=k+1}^{n} e_i(Y_i) \mid A, \mathcal{F}_k\right] \mid A, \mathcal{F}_{k-1}\right] \pi_k(X_k)$$

$$= \mathbb{E}\left[e_k(\mu_k(X_k)) \mathbb{E}\left[\prod_{i=k+1}^{n} e_i(Y_i) \mid A, \mathcal{F}_k\right] \mid A, \mathcal{F}_{k-1}\right]$$

$$\quad + \mathbb{E}\left[\left(e_k(Y_k) - e_k(\mu_k(X_k))\right) \mathbb{E}\left[\prod_{i=k+1}^{n} e_i(Y_i) \mid A, \mathcal{F}_k\right] \mid A, \mathcal{F}_{k-1}\right]$$

$$= \mathbb{E}\left[e_k(Y_k) \mathbb{E}\left[\prod_{i=k+1}^{n} e_i(Y_i) \mid A, \mathcal{F}_k\right] \mid A, \mathcal{F}_{k-1}\right] = \mathbb{E}\left[\mathbb{E}\left[e_k(Y_k) \prod_{i=k+1}^{n} e_i(Y_i) \mid A, \mathcal{F}_k\right] \mid A, \mathcal{F}_{k-1}\right]$$

$$= \mathbb{E}\left[\mathbb{E}\left[\prod_{i=k}^{n} e_i(Y_i) \mid A, \mathcal{F}_k\right] \mid A, \mathcal{F}_{k-1}\right] = \mathbb{E}\left[\prod_{i=k}^{n} e_i(Y_i) \mid A, \mathcal{F}_{k-1}\right],$$

as we desired. $\qquad\square$

By picking $k = 1$ and $A$ to be a trivial event in Lemma A.2, we conclude that $\mathbb{E}[E_n^{\mathrm{ppi}}] = \mathbb{E}[\prod_{i=1}^{n} e_i^{\mathrm{ppi}} \mid \mathcal{F}_0] = \mathbb{E}[\prod_{i=1}^{n} e_i(Y_i) \mid \mathcal{F}_0] = \mathbb{E}[E_n] \leq 1$, and so $E_n^{\mathrm{ppi}}$ is a valid e-value.

Now let us show that, if the underlying e-values form a test supermartingale, then so is the prediction-powered process. By definition $E_0^{\mathrm{ppi}} = E_0 = 1$, and so all we need to do is to show that $\mathbb{E}[E_{n+1}^{\mathrm{ppi}} \mid \mathcal{F}_n] \leq E_n^{\mathrm{ppi}}$. It follows:

$$\mathbb{E}[E_{n+1}^{\mathrm{ppi}} \mid \mathcal{F}_n] = \mathbb{E}[e_{n+1}^{\mathrm{ppi}} \cdot E_n^{\mathrm{ppi}} \mid \mathcal{F}_n] = \mathbb{E}[e_{n+1}^{\mathrm{ppi}} \mid \mathcal{F}_n] \cdot E_n^{\mathrm{ppi}}$$

$$= \mathbb{E}\left[e_{n+1}(\mu_{n+1}(X_{n+1})) + \frac{\xi_{n+1}}{\pi_{n+1}(X_{n+1})}\left(e_{n+1}(Y_{n+1}) - e_{n+1}(\mu_{n+1}(X_{n+1}))\right) \mid \mathcal{F}_n\right] \cdot E_n^{\mathrm{ppi}}$$

$$= \mathbb{E}\left[e_{n+1}(\mu_{n+1}(X_{n+1})) \mid \mathcal{F}_n\right] \cdot E_n^{\mathrm{ppi}}$$

$$+ \mathbb{E}\left[\frac{\xi_{n+1}}{\pi_{n+1}(X_{n+1})}\left(e_{n+1}(Y_{n+1}) - e_{n+1}(\mu_{n+1}(X_{n+1}))\right) \mid \xi_{n+1} = 1, \mathcal{F}_n\right]\mathbb{P}[\xi_{n+1} = 1 \mid \mathcal{F}_n] \cdot E_n^{\mathrm{ppi}}$$

$$+ \mathbb{E}\left[\frac{\xi_{n+1}}{\pi_{n+1}(X_{n+1})}\left(e_{n+1}(Y_{n+1}) - e_{n+1}(\mu_{n+1}(X_{n+1}))\right) \mid \xi_{n+1} = 0, \mathcal{F}_n\right]\mathbb{P}[\xi_{n+1} = 0 \mid \mathcal{F}_n] \cdot E_n^{\mathrm{ppi}}$$

$$= \mathbb{E}\left[e_{n+1}(\mu_{n+1}(X_{n+1})) \mid \mathcal{F}_n\right] \cdot E_n^{\mathrm{ppi}}$$

$$+ \mathbb{E}\left[\frac{1}{\pi_{n+1}(X_{n+1})}\left(e_{n+1}(Y_{n+1}) - e_{n+1}(\mu_{n+1}(X_{n+1}))\right) \mid \xi_{n+1} = 1, \mathcal{F}_n\right]\pi_{n+1}(X_{n+1}) \cdot E_n^{\mathrm{ppi}}$$

$$= \mathbb{E}\left[e_{n+1}(\mu_{n+1}(X_{n+1})) \mid \mathcal{F}_n\right] \cdot E_n^{\mathrm{ppi}} + \mathbb{E}\left[e_{n+1}(Y_{n+1}) - e_{n+1}(\mu_{n+1}(X_{n+1})) \mid \xi_{n+1} = 1, \mathcal{F}_n\right] \cdot E_n^{\mathrm{ppi}}$$

$$= \mathbb{E}\left[e_{n+1}(Y_{n+1}) \mid \mathcal{F}_n\right] \cdot E_n^{\mathrm{ppi}}$$

$$= \mathbb{E}\left[e_{n+1}(Y_{n+1}) \mid \mathcal{F}_n\right] \cdot E_n \cdot \frac{E_n^{\mathrm{ppi}}}{E_n}$$

$$\leq E_n \cdot \frac{E_n^{\mathrm{ppi}}}{E_n} = E_n^{\mathrm{ppi}}.$$

Finally, we assume that the underlying e-values form an e-process for finite stopping times. We want to show that, for any finite stopping time $\tau$, $\mathbb{E}[E_\tau^{\mathrm{ppi}}] \leq 1$. Well,

$$\mathbb{E}[E_\tau^{\mathrm{ppi}}] = \mathbb{E}[\mathbb{E}[E_\tau^{\mathrm{ppi}} \mid \tau]];$$

When $\tau = n$ for each $n \in \mathbb{N}$, by Lemma A.2 with $k = n$ and $A = \{\tau = n\}$, it holds that $\mathbb{E}[E_n^{\mathrm{ppi}} \mid \tau = n] = \mathbb{E}[\prod_{i=1}^n e_i^{\mathrm{ppi}} \mid \tau = n, \mathcal{F}_0] = \mathbb{E}[\prod_{i=1}^n e_i(Y_i) \mid \tau = n, \mathcal{F}_0] = \mathbb{E}[E_n \mid \tau = n]$.

Thus

$$\mathbb{E}[E_\tau^{\mathrm{ppi}}] = \mathbb{E}[\mathbb{E}[E_\tau^{\mathrm{ppi}} \mid \tau]] = \mathbb{E}[\mathbb{E}[E_\tau \mid \tau]] = \mathbb{E}[E_\tau] \leq 1,$$

and so $E^{\mathrm{ppi}}$ is an e-process. $\qquad\square$

To prove the main theorem about power of our prediction-powered e-values, we will use the following change-of-measure lemma based on the Wasserstein distance:

**Lemma A.3.** *For any distributions $P$ and $Q$ over some space $\mathcal{Z}$ and any $L$-Lipschitz function $\phi : \mathcal{Z} \to \mathbb{R}$,*

$$|\mathbb{E}_P[\phi] - \mathbb{E}_Q[\phi]| \leq L\, W(P\|Q),$$

*where $W(P\|Q)$ is the Wasserstein distance between $P$ and $Q$.*

*Proof.* The proof follows immediately from the representation of the Wasserstein distance as an IPM. The Wasserstein distance, written as an IPM, is

$$W(P\|Q) = \sup_{\|f\|_{\mathrm{Lip}}=1} |\mathbb{E}_P[f] - \mathbb{E}_Q[f]|.$$

If $\phi$ is $L$-Lipschitz, then $\phi/L$ is 1-Lipschitz, and so

$$|\mathbb{E}_P[\phi] - \mathbb{E}_Q[\phi]| = |L\mathbb{E}_P[\phi/L] - L\mathbb{E}_Q[\phi/L]| = L|\mathbb{E}_P[\phi/L] - \mathbb{E}_Q[\phi/L]| \leq L \sup_{\|f\|_{\mathrm{Lip}}=1} |\mathbb{E}_P[f] - \mathbb{E}_Q[f]| = L\, W(P\|Q).$$

$\qquad\square$

We will also use a simple upper bound on the Wasserstein distance, showing that it is upper bounded by the MAE.

**Lemma A.4.** *For any distributions $P$ and $Q$ over some normed space $(\mathcal{Z}, \|\cdot\|)$,*

$$W(P, Q) \leq \mathbb{E}_{Z_P \sim P, Z_Q \sim Q}[\|Z_P - Z_Q\|].$$

*Proof.* By the dual representation of the Wasserstein distance,

$$
\begin{aligned}
W(P\|Q) &= \sup_{\|f\|_{\mathrm{Lip}}=1} |\mathbb{E}_{Z_P \sim P}[f(Z_P)] - \mathbb{E}_{Z_Q \sim Q}[f(Z_Q)]| = \sup_{\|f\|_{\mathrm{Lip}}=1} |\mathbb{E}_{Z_P \sim P, Z_Q \sim Q}[f(Z_P) - f(Z_Q)]| \\
&\leq \sup_{\|f\|_{\mathrm{Lip}}=1} \mathbb{E}_{Z_P \sim P, Z_Q \sim Q}[|f(Z_P) - f(Z_Q)|] \leq \sup_{\|f\|_{\mathrm{Lip}}=1} \mathbb{E}_{Z_P \sim P, Z_Q \sim Q}[\|Z_P - Z_Q\|] \\
&= \mathbb{E}_{Z_P \sim P, Z_Q \sim Q}[\|Z_P - Z_Q\|].
\end{aligned}
$$

$\square$

**Theorem A.5** (Theorem 2.2 in the main text)**.** *Suppose that the $e_i(\cdot)$ are each $L_i$-Lipschitz, and that $\pi_i(X_i) \geq 1 - a_i/b_i + \epsilon_i$ for some $\epsilon_i > 0$, for all $i$. Then there exists some constant $c > 0$ independent of $n$ such that*

$$\mathbb{E}\left[\frac{1}{n}\log E_n^{\mathrm{ppi}}\right] \geq \mathbb{E}\left[\frac{1}{n}\log E_n\right] - \frac{c}{n}\sum_{i=1}^n \mathbb{E}[\|\mu_i(X_i) - Y_i\|],$$

*Proof.* First, note that

$$
\begin{aligned}
\mathbb{E}\left[\frac{1}{n}\log E_n^{\mathrm{ppi}}\right] &= \mathbb{E}\left[\frac{1}{n}\log \prod_{i=1}^n e_i^{\mathrm{ppi}}\right] = \frac{1}{n}\sum_{i=1}^n \mathbb{E}\left[\log e_i^{\mathrm{ppi}}\right] \\
&= \frac{1}{n}\sum_{i=1}^n \mathbb{E}\left[\log\left(e_i(\mu_i(X_i)) + \frac{\xi_i}{\pi_i(X_i)}[e_i(Y_i) - e_i(\mu_i(X_i))]\right)\right]. \\
&= \frac{1}{n}\sum_{i=1}^n \mathbb{E}\left[\mathbb{E}\left[\log\left(e_i(\mu_i(X_i)) + \frac{\xi_i}{\pi_i(X_i)}[e_i(Y_i) - e_i(\mu_i(X_i))]\right) \mid Y_i, \xi_i, \pi_i(X_i), \mathcal{F}_{i-1}\right]\right].
\end{aligned}
$$

The inner expectation in the last line is random only over $\mu_i(X_i)$. Moreover, thanks to our assumptions, the value we are taking the expectation over is Lipschitz as a function of $\mu_i(X_i)$: because of the lower bound on the $\pi_i(X_i)$ with positive margins $\epsilon_i$, the value within the log is bounded away from zero, and so the log becomes Lipschitz with some constant $u > 0$.

$$\left\|\log\left(e_i(\cdot) + \frac{\xi_i}{\pi_i(X_i)}[e_i(Y_i) - e_i(\cdot)]\right)\right\|_{\mathrm{Lip}} \leq u\left\|e_i(\cdot) + \frac{\xi_i}{\pi_i(X_i)}[e_i(Y_i) - e_i(\cdot)]\right\|_{\mathrm{Lip}};$$

If $\xi_i = 0$, then this equals $u\|e_i(\cdot)\|_{\mathrm{Lip}} = u \cdot L_i$. Otherwise, this equals

$$
\begin{aligned}
u\left\|e_i(\cdot) + \frac{\xi_i}{\pi_i(X_i)}[e_i(Y_i) - e_i(\cdot)]\right\|_{\mathrm{Lip}} &= u\left\|\frac{e_i(Y_i) - (1 - \pi_i(X_i))e_i(\cdot)}{\pi_i(X_i)}\right\|_{\mathrm{Lip}} \\
&= u\frac{1}{\pi_i(X_i)}\|e_i(Y_i) - (1 - \pi_i(X_i))e_i(\cdot)\|_{\mathrm{Lip}} \\
&= u\frac{1}{\pi_i(X_i)}\|-(1 - \pi_i(X_i))e_i(\cdot)\|_{\mathrm{Lip}} \\
&= u\frac{(1 - \pi_i(X_i))}{\pi_i(X_i)}\|e_i(\cdot)\|_{\mathrm{Lip}} = u \cdot L_i \cdot \frac{(1 - \pi_i(X_i))}{\pi_i(X_i)}.
\end{aligned}
$$

In either case, this Lipschitz constant is upper bounded by $c := u \cdot L_i \cdot \max\left\{\frac{(1 - \pi_i(X_i))}{\pi_i(X_i)}, 1\right\}$ (which does not depend on $n$).

Hence, by Lemma A.3,

$$\frac{1}{n} \sum_{i=1}^{n} \mathbb{E} \left[ \mathbb{E} \left[ \log \left( e_i(\mu_i(X_i)) + \frac{\xi_i}{\pi_i(X_i)} [e_i(Y_i) - e_i(\mu_i(X_i))] \right) \mid Y_i, \xi_i, \pi_i(X_i), \mathcal{F}_{i-1} \right] \right]$$

$$\geq \frac{1}{n} \sum_{i=1}^{n} \mathbb{E} \left[ \mathbb{E} \left[ \log \left( e_i(Y_i) + \frac{\xi_i}{\pi_i(X_i)} [e_i(Y_i) - e_i(Y_i)] \right) \mid Y_i, \xi_i, \pi_i(X_i), \mathcal{F}_{i-1} \right] - c \, W(\mu_i(X_i) \| Y_i) \right]$$

$$= \frac{1}{n} \sum_{i=1}^{n} \mathbb{E} \left[ \mathbb{E} \left[ \log e_i(Y_i) \mid Y_i, \xi_i, \pi_i(X_i), \mathcal{F}_{i-1} \right] - c \, W(\mu_i(X_i) \| Y_i) \right]$$

$$= \frac{1}{n} \sum_{i=1}^{n} \mathbb{E} \left[ \log e_i(Y_i) \right] - \frac{1}{n} \sum_{i=1}^{n} \mathbb{E} \left[ c \, W(\mu_i(X_i) \| Y_i) \right]$$

$$= \mathbb{E} \left[ \frac{1}{n} \log E_n \right] - \frac{c}{n} \sum_{i=1}^{n} \mathbb{E} \left[ W(\mu_i(X_i) \| Y_i) \right].$$

Apply Lemma A.4 and conclude. $\qquad \square$

The following is a more precise statement about the growth rate of our prediction-powered e-values, albeit less directly interpretable:

**Theorem A.6.** *It holds that*

$$\mathbb{E} \left[ \frac{1}{n} \log E_n^{\mathrm{ppi}} \right] = \mathbb{E} \left[ \frac{1}{n} \sum_{i=1}^{n} (1 - \pi_i(X_i)) \log e_i(\mu_i(X_i)) \right] + \mathbb{E} \left[ \frac{1}{n} \sum_{i=1}^{n} \pi_i(X_i) \log \left( e_i(\mu_i(X_i)) + \frac{e_i(Y_i) - e_i(\mu_i(X_i))}{\pi_i(X_i)} \right) \right].$$

*Proof.*

$$\mathbb{E} \left[ \frac{1}{n} \log E_n^{\mathrm{ppi}} \right] = \mathbb{E} \left[ \frac{1}{n} \log \prod_{i=1}^{n} e_i^{\mathrm{ppi}} \right] = \frac{1}{n} \sum_{i=1}^{n} \mathbb{E} \left[ \log e_i^{\mathrm{ppi}} \right]$$

$$= \frac{1}{n} \sum_{i=1}^{n} \mathbb{E} \left[ \log \left( e_i(\mu_i(X_i)) + [e_i(Y_i) - e_i(\mu_i(X_i))] \cdot \frac{\xi_i}{\pi_i(X_i)} \right) \right]$$

$$= \frac{1}{n} \sum_{i=1}^{n} \mathbb{E} \left[ \mathbb{E} \left[ \log \left( e_i(\mu_i(X_i)) + [e_i(Y_i) - e_i(\mu_i(X_i))] \cdot \frac{\xi_i}{\pi_i(X_i)} \right) \mid \mathcal{F}_{i-1} \right] \right]$$

$$= \frac{1}{n} \sum_{i=1}^{n} \mathbb{E} \left[ \mathbb{E} \left[ \log \left( e_i(\mu_i(X_i)) + [e_i(Y_i) - e_i(\mu_i(X_i))] \cdot \frac{\xi_i}{\pi_i(X_i)} \right) \mid \xi_i = 1, \mathcal{F}_{i-1} \right] \mathbb{P}[\xi_i = 1 \mid \mathcal{F}_{i-1}] \right.$$

$$\left. + \mathbb{E} \left[ \log \left( e_i(\mu_i(X_i)) + [e_i(Y_i) - e_i(\mu_i(X_i))] \cdot \frac{\xi_i}{\pi_i(X_i)} \right) \mid \xi_i = 0, \mathcal{F}_{i-1} \right] \mathbb{P}[\xi_i = 0 \mid \mathcal{F}_{i-1}] \right]$$

$$= \frac{1}{n} \sum_{i=1}^{n} \mathbb{E} \left[ \mathbb{E} \left[ \log \left( e_i(\mu_i(X_i)) + [e_i(Y_i) - e_i(\mu_i(X_i))] \cdot \frac{1}{\pi_i(X_i)} \right) \mid \mathcal{F}_{i-1} \right] \pi_i(X_i) \right.$$

$$\left. + \mathbb{E} \left[ \log e_i(\mu_i(X_i)) \mid \mathcal{F}_{i-1} \right] (1 - \pi_i(X_i)) \right]$$

$$= \frac{1}{n} \sum_{i=1}^{n} \mathbb{E} \left[ \pi_i(X_i) \log \left( e_i(\mu_i(X_i)) + [e_i(Y_i) - e_i(\mu_i(X_i))] \cdot \frac{1}{\pi_i(X_i)} \right) + (1 - \pi_i(X_i)) \log e_i(\mu_i(X_i)) \right]$$

$$= \mathbb{E} \left[ \frac{1}{n} \sum_{i=1}^{n} (1 - \pi_i(X_i)) \log e_i(\mu_i(X_i)) \right] + \mathbb{E} \left[ \frac{1}{n} \sum_{i=1}^{n} \pi_i(X_i) \log \left( e_i(\mu_i(X_i)) + \frac{e_i(Y_i) - e_i(\mu_i(X_i))}{\pi_i(X_i)} \right) \right].$$

$$\square$$

To prove the next result we will make use of Ville's inequality:

**Theorem A.7** (Ville's inequality (Ville, 1939; Ramdas, 2018)). *For any nonnegative supermartingale $(L_t)$ and any $x > 1$, define the (possibly infinite) stopping time $N := \inf t \geq 1 : L_t \geq x$ and denote the expected overshoot when $L_t$ surpasses $x$ as*

$$o = \mathbb{E}\left[\frac{L_N}{x} \mid N < \infty\right] \geq 1.$$

*Then,*

$$\mathbb{P}[\exists t : L_t \geq x] \leq \frac{\mathbb{E}[L_0]}{ox} \geq \frac{\mathbb{E}[L_0]}{x}.$$

**Proposition A.8** (Proposition 2.3 in the main text). $C_n^{\text{ppi}-(\alpha)}$ *is a valid confidence interval – i.e., $\mathbb{P}[\theta^\star \in C_n^{\text{ppi}-(\alpha)}] \geq 1 - \alpha$. Moreover:*

 (i) *If the underlying e-values form a nonnegative supermartingale, then the prediction-powered intervals are anytime-valid (also known as confidence sequences): $\mathbb{P}[\forall n \in \mathbb{N}, \ \theta^\star \in C_n^{\text{ppi}-(\alpha)}] \geq 1 - \alpha$;*

 (ii) *More generally, if the underlying e-values form e-processes, then the prediction-powered intervals are valid at arbitrary stopping times: $\mathbb{P}[\theta^\star \in C_\tau^{\text{ppi}-(\alpha)}] \geq 1 - \alpha$ for any stopping time $\tau$.*

*Proof.* By construction, $\mathbb{P}[\theta^\star \in C_n^{\text{ppi}-(\alpha)}] = \mathbb{P}[E_n^{\text{ppi}-(\theta^\star)} \leq 1/\alpha] = 1 - \mathbb{P}[E_n^{\text{ppi}-(\theta^\star)} > 1/\alpha]$. By Markov, considering that the null $H_0^{\theta^\star}$ holds and using Theorem 2.1,

$$1 - \mathbb{P}[E_n^{\text{ppi}-(\theta^\star)} > 1/\alpha] \geq 1 - \frac{\mathbb{E}[E_n^{\text{ppi}-(\theta^\star)}]}{1/\alpha} \geq 1 - \frac{1}{1/\alpha} = 1 - \alpha.$$

If the underlying e-values form a test supermartingale, then by Theorem 2.1 so do the prediction-powered e-values; then, using Ville's inequality,

$$\mathbb{P}[\forall n \in \mathbb{N}, \ \theta^\star \in C_n^{\text{ppi}-(\alpha)}] = \mathbb{P}[\forall n \in \mathbb{N}, \ E_n^{\text{ppi}-(\theta^\star)} \leq 1/\alpha] = \mathbb{P}[\sup_n E_n^{\text{ppi}-(\theta^\star)} \leq 1/\alpha]$$

$$= 1 - \mathbb{P}[\sup_n E_n^{\text{ppi}-(\theta^\star)} > 1/\alpha] \geq 1 - \frac{\mathbb{E}[E_0^{\text{ppi}-(\theta^\star)}]}{1/\alpha} = 1 - \frac{1}{1/\alpha} = 1 - \alpha.$$

Finally, if the underlying e-values form an e-process, thenby Theorem 2.1 so do the prediction-powered e-values (for finite stopping times), and so, by Markov,

$$\mathbb{P}[\theta^\star \in C_\tau^{\text{ppi}-(\alpha)}] = \mathbb{P}[E_\tau^{\text{ppi}-(\theta^\star)} \leq 1/\alpha] = 1 - \mathbb{P}[E_\tau^{\text{ppi}-(\theta^\star)} > 1/\alpha]$$

$$\geq 1 - \frac{\mathbb{E}[E_\tau^{\text{ppi}-(\theta^\star)}]}{1/\alpha} \geq 1 - \frac{1}{1/\alpha} = 1 - \alpha.$$

$\square$

**Proposition A.9** (Proposition 2.4 in the main text). *Under the assumptions of Theorem 2.2, let $\nu$ be a measure over the parameter space $\Theta$. Then there exists some $c$ for which*

$$\mathbb{E}\left[\int \frac{1}{n} \log \frac{1}{E_n^{\text{ppi}-(\theta)}} \mathrm{d}\nu(\theta)\right] \leq \mathbb{E}\left[\int \frac{1}{n} \log \frac{1}{E_n^{(\theta)}} \mathrm{d}\nu(\theta)\right] + \frac{\nu(\Theta)c}{n} \sum_{i=1}^{n} \mathbb{E}[\|\mu_i(X_i) - Y_i\|].$$

*Proof.* By Fubini,

$$\mathbb{E}\left[\int \frac{1}{n} \log 1/E_n^{\text{ppi}-(\theta)} \mathrm{d}\nu(\theta)\right] = \int \mathbb{E}\left[\frac{1}{n} \log 1/E_n^{\text{ppi}-(\theta)}\right] \mathrm{d}\nu(\theta)$$

And now we apply Theorem 2.2:

$$\int \mathbb{E}\left[\frac{1}{n}\log 1/E_n^{\mathrm{ppi}-(\theta)}\right]\mathrm{d}\nu(\theta) = -\int \mathbb{E}\left[\frac{1}{n}\log E_n^{\mathrm{ppi}-(\theta)}\right]\mathrm{d}\nu(\theta) \le -\int\left(\mathbb{E}\left[\frac{1}{n}\log E_n^{(\theta)}\right] - \frac{c}{n}\sum_{i=1}^n \mathbb{E}[W(\mu_i(X_i)\|Y_i)]\right)\mathrm{d}\nu(\theta)$$

$$\le \int\left(\mathbb{E}\left[\frac{1}{n}\log 1/E_n^{(\theta)}\right] + \frac{c}{n}\sum_{i=1}^n \mathbb{E}[W(\mu_i(X_i)\|Y_i)]\right)\mathrm{d}\nu(\theta)$$

$$= \int \mathbb{E}\left[\frac{1}{n}\log 1/E_n^{(\theta)}\right]\mathrm{d}\nu(\theta) + \frac{\nu(\Theta)c}{n}\sum_{i=1}^n \mathbb{E}[W(\mu_i(X_i)\|Y_i)]$$

$$= \mathbb{E}\left[\int \frac{1}{n}\log 1/E_n^{(\theta)}\mathrm{d}\nu(\theta)\right] + \frac{\nu(\Theta)c}{n}\sum_{i=1}^n \mathbb{E}[W(\mu_i(X_i)\|Y_i)]$$

$$\le \mathbb{E}\left[\int \frac{1}{n}\log 1/E_n^{(\theta)}\mathrm{d}\nu(\theta)\right] + \frac{\nu(\Theta)c}{n}\sum_{i=1}^n \mathbb{E}[\|\mu_i(X_i) - Y_i\|],$$

where the last step holds by Lemma A.4. $\qquad\square$

Considering that the object of interest is a confidence interval, it is desirable to further bound the *measure* of the interval. We were unable to prove any sufficiently general result that was (i) nonvacuous, and (ii) decayed reasonably fast as $n$ increased, and imagine that heavy assumptions are necessary; this may be best done on a case-by-case basis. Nevertheless, here is one possible somewhat straightforward result.

**Proposition A.10.** *Under the same conditions of Proposition 2.4, suppose that the prediction-powered e-values are bounded from above by $M^{\mathrm{ppi}}$ (i.e., for all $\theta \in \Theta$, $E_n^{\mathrm{ppi}-(\theta)} < M^{\mathrm{ppi}}$ almost surely), and similarly for the non-prediction powered e-values by $M$ (i.e., for all $\theta \in \Theta$, $E_n^{(\theta)} < M$ almost surely). Then: Then*

$$\mathbb{E}[\nu(C^{\mathrm{ppi}})] \le \frac{\mathbb{E}[\int \log 1/E^{\mathrm{ppi}-(\theta)}\mathrm{d}\nu(\theta)] + \nu(\Theta)M^{\mathrm{ppi}}}{\log\alpha + \log M^{\mathrm{ppi}}}, \qquad \mathbb{E}[\nu(C)] \le \frac{\mathbb{E}[\int \log 1/E^{(\theta)}\mathrm{d}\nu(\theta)] + \nu(\Theta)M}{\log\alpha + \log M}.$$

*Proof.* Consider the measure $\tilde{\nu}(A) = \nu(A)/\nu(\Theta)$; it is a probability measure. Then:

$$\tilde{\nu}(C^{\mathrm{ppi}}) = \mathbb{P}_{\theta\sim\tilde{\nu}}[E^{\mathrm{ppi}-(\theta)} < 1/\alpha] = \mathbb{P}_{\theta\sim\tilde{\nu}}[1/E^{\mathrm{ppi}-(\theta)} > \alpha] = \mathbb{P}_{\theta\sim\tilde{\nu}}[\log 1/E^{\mathrm{ppi}-(\theta)} > \log\alpha];$$

We want to apply Markov. To do that, we need the left-hand side to be nonnegative; to do so, we add $\log M^{\mathrm{ppi}}$ to both sides, which yields

$$\mathbb{P}_{\theta\sim\tilde{\nu}}[\log 1/E^{\mathrm{ppi}-(\theta)} > \log\alpha]; = \mathbb{P}_{\theta\sim\tilde{\nu}}[\log 1/E^{\mathrm{ppi}-(\theta)} + \log M^{\mathrm{ppi}} > \log\alpha + \log M^{\mathrm{ppi}}]$$

$$\le \frac{\mathbb{E}_{\theta\sim\tilde{\nu}}[\log 1/E^{\mathrm{ppi}-(\theta)} + \log M^{\mathrm{ppi}}]}{\log\alpha + \log M^{\mathrm{ppi}}}$$

$$\le \frac{\int \log 1/E^{\mathrm{ppi}-(\theta)}\mathrm{d}\tilde{\nu}(\theta) + \log M^{\mathrm{ppi}}}{\log\alpha + \log M^{\mathrm{ppi}}}.$$

$$\le \frac{[\nu(\Theta)]^{-1}\int \log 1/E^{\mathrm{ppi}-(\theta)}\mathrm{d}\nu(\theta) + \log M^{\mathrm{ppi}}}{\log\alpha + \log M^{\mathrm{ppi}}}.$$

So, multiplying everything by $\nu(\Theta)$, we get that

$$\tilde{\nu}(C^{\mathrm{ppi}})\cdot\nu(\Theta) = \nu(C^{\mathrm{ppi}}) \le \nu(\Theta)\cdot\frac{[\nu(\Theta)]^{-1}\int \log 1/E^{\mathrm{ppi}-(\theta)}\mathrm{d}\nu(\theta) + \log M^{\mathrm{ppi}}}{\log\alpha + \log M^{\mathrm{ppi}}} = \frac{\int \log 1/E^{\mathrm{ppi}-(\theta)}\mathrm{d}\nu(\theta) + \nu(\Theta)\log M^{\mathrm{ppi}}}{\log\alpha + \log M^{\mathrm{ppi}}}.$$

Finally, taking the expectation on both sides, we get that

$$\mathbb{E}[\tilde{\nu}(C^{\mathrm{ppi}})] \le \mathbb{E}\left[\frac{\int \log 1/E^{\mathrm{ppi}-(\theta)}\mathrm{d}\nu(\theta) + \nu(\Theta)\log M^{\mathrm{ppi}}}{\log\alpha + \log M^{\mathrm{ppi}}}\right] = \frac{\mathbb{E}[\int \log 1/E^{\mathrm{ppi}-(\theta)}\mathrm{d}\nu(\theta)] + \nu(\Theta)\log M^{\mathrm{ppi}}}{\log\alpha + \log M^{\mathrm{ppi}}},$$

as we desired.

The same can be done for the non-prediction-powered e-values, replacing $E^{\mathrm{ppi}}$ with $E$ and $M^{\mathrm{ppi}}$ with $M$. $\qquad\square$

Most terms in the inequality depend on $n$, so it's a bit hard to intuit. But, if the dependence on the $n$ in the expectation of the log is good enough, then this should be nonvacuous, at least.

**Proposition A.11** (Proposition 2.6 in the main text). *Under Assumption 2.5, it holds that* $\mathcal{A}((E_n^{\text{ppi}-(\gamma)})_{\gamma\in\Gamma})$ *is also* valid. *If the underlying e-values are e-processes, then it further holds that* $\mathcal{A}((E_\tau^{\text{ppi}-(\gamma)})_{\gamma\in\Gamma})$ *is* valid *for any finite stopping time* $\tau$.

*Proof.* To prove that $\mathcal{A}((E_n^{\text{ppi}-(\gamma)})_{\gamma\in\Gamma})$ is valid, by Assumption 2.5, it suffices to show that $E_n^{\text{ppi}-(\gamma)}$ is valid for every $\gamma \in \Gamma$; and by Theorem 2.1, this is indeed the case.

Now suppose that the underlying e-values $(E_n^{(\gamma)})_{\gamma\in\Gamma}$ form e-processes; then so do the prediction-powered e-values $(E_n^{\text{ppi}-(\gamma)})_{\gamma\in\Gamma}$ for finite stopping times, by Theorem 2.1. Then, to prove that $\mathcal{A}((E_\tau^{\text{ppi}-(\gamma)})_{\gamma\in\Gamma})$ is valid for any finite stopping time $\tau$, again by Assumption 2.5 it suffices to show that $E_\tau^{\text{ppi}-(\gamma)}$ is valid, which is indeed the case since they form e-processes for finite stopping times. □

## B. Additional Results

### B.1. The Asymptotic Setting

E-values, though usually defined in non-asymptotic terms, have asymptotic analogues. In particular, a (sequential) asymptotic e-value is defined as a (sequence of) nonnegative random variable(s) $E_n$ such that, under the null $H_0$, it holds that $\limsup_{n\to\infty} \mathbb{E}[E_n] \leq 1$ (Ramdas & Wang, 2024). We briefly show here that the core points of the theory we build in the main text can be directly applied here. Most results whose analogues we do not prove still hold, and are just omitted for conciseness.

**Proposition B.1.** *If $E_n$ is an asymptotic e-value, then so is its prediction-powered analogue $E_n^{\text{ppi}}$.*

*Proof.* We want to prove that $E_n^{\text{ppi}}$ is an asymptotic e-value. It follows, by Theorem 2.1:

$$\limsup_{n\to\infty} \mathbb{E}[E_n^{\text{ppi}}] = \limsup_{n\to\infty} \mathbb{E}[E_n] \leq 1.$$

□

**Proposition B.2.** *If $E_n^{(\theta)}$ is an asymptotic e-value for each $\theta in \Theta$, then $C_n^{\text{ppi}-(\alpha)} := \{\theta \in \Theta : E_n^{\text{ppi}-(\theta)} < 1/\alpha\}$ is an asymptotic confidence interval, i.e.,* $\limsup_{n\to\infty} \mathbb{P}[\theta^\star \notin C_n^{\text{ppi}-(\alpha)}] \leq \alpha$.

*Proof.* It holds that

$$\limsup_{n\to\infty} \mathbb{P}[\theta^\star \notin C_n^{\text{ppi}-(\alpha)}] = \limsup_{n\to\infty} \mathbb{P}[E_n^{\text{ppi}-(\theta^\star)} \geq 1/\alpha];$$

By Markov,

$$\limsup_{n\to\infty} \mathbb{P}[E_n^{\text{ppi}-(\theta^\star)} \geq 1/\alpha] \leq \limsup_{n\to\infty} \frac{\mathbb{E}[E_n^{\text{ppi}-(\theta^\star)}]}{1/\alpha} = \alpha \limsup_{n\to\infty} \mathbb{E}[E_n^{\text{ppi}-(\theta^\star)}] \leq \alpha.$$

□

The results related to power (e.g., Theorem 2.2) apply to asymptotic e-values without any modification necessary.

### B.2. An approximately optimal choice for $\pi_i$

Our prediction-powered e-values have, at their core, the customizeable choice of data collection probabilities $\pi_i(X_i)$. While selecting a constant $\pi_i(X_i) = \pi_{\text{inf}}$, where $\pi_{\text{inf}}$ is the lowest possible value possible (so as to minimize data collection costs) is a reasonable approach, it ignores the versatility that the probability can take into account the 'cheap' data $X_i$, which could significantly improve statistical power when used correctly. In an effort to seek a better strategy, we try to identify an 'approximately optimal' choice of $\pi_i$.

The optimality is in the sense that, at point $i$ in time, the data collection probability function $\pi_i(\cdot)$ should be chosen so as to maximize the expected log of the e-value, as per (Kelly, 1956); this is also similar, e.g., to the GRAPA and aGRAPA strategies of (Waudby-Smith & Ramdas, 2020). However, the $\pi$ also have additional constraints:

(i) Its image must be bounded: $\pi_i : \mathcal{X} \to [1 - a_i/b_i, 1]$. I.e., for all $x \in \mathcal{X}$, $1 - a_i/b_i \leq \pi_i(x) \leq 1$.

(ii) It must respect some particular maximal data collection budget: $\mathbb{E}[\pi_i(X_i)] \leq \text{Budget}$.

So we seek to solve the following constrained functional optimization problem:

$$\pi_i^{\star} = \underset{\pi \in L^2}{\text{argmax}} \, \mathbb{E}[\log E_n^{\text{ppi}} \mid \mathcal{F}_{i-1}] = \underset{\pi \in L^2}{\text{argmin}} \, \mathbb{E}[-\log e_n^{\text{ppi}} \mid \mathcal{F}_{i-1}]$$

subject to

$$1 - a_i/b_i \leq \pi_i(x) \leq 1 \quad \text{for (almost) all } x \in \mathcal{X}$$
$$\mathbb{E}[\pi(X_i) \mid \mathcal{F}_{i-1}] \leq \text{Budget},$$

where we assume that the domain of $\pi$ is bounded (so that there are functions that satisfy the first domain, since $\pi$ is always positive).

Our approximate solution to this is as follows: the functional gradient of our (unconstrained) loss is given by

$$\pi \mapsto \mathbb{E}\left[h\left(\frac{e_i(Y_i) - (1 - \pi(X_i))e_i(\mu_i(X_i))}{e_i(\mu_i(X_i)) \cdot \pi(X_i)} - \log\right) - 1 \mid X_i, \mathcal{F}_{i-1}\right],$$

where $h(t) = 1/t - \log 1/t = 1/t + \log t$. The $h$ function is a bit inconvenient for solving this problem in closed form, so, inspired by (Waudby-Smith & Ramdas, 2020), we do a Taylor approximation around some point $a$ (which turns out to later combine with the parameter to control the budget constraint); this leads to the following approximate functional gradient:

$$\pi \mapsto \alpha_a + \beta_a/\pi_{\text{inf}}\mathbb{E}\left[\frac{e_i(Y_i)}{e_i(\mu_i(X_i))} \mid X_i, \mathcal{F}_{i-1}\right] - \beta_a \frac{1 - \pi_{\text{inf}}}{\pi_{\text{inf}}},$$

where $\alpha_a = \log a + 2/a - 2$ and $\beta_a = (a - 1)/a^2$.

The unconstrained solution is then given by

$$\pi^{\star}(X_i) \approx -\left(\mathbb{E}\left[\frac{e_i(Y_i)}{e_i(\mu_i(X_i))} \mid X_i, \mathcal{F}_{i-1}\right] - 1\right)/(\alpha_a/\beta_a + 1),$$

and KKT conditions give that:

- If the unconstrained optimum above satisfies the boundedness constraint, then that is the optimal choice;

- If $\alpha_a + \beta_a(\mathbb{E}\left[\frac{e_i(Y_i)}{e_i(\mu_i(X_i))} \mid X_i, \mathcal{F}_{i-1}\right]/\pi_{\text{inf}} - (1 - \pi_{\text{inf}})/\pi_{\text{inf}}) \leq 0$, then $\pi^{\star}(X_i) = \pi_{\text{inf}}$;

- Otherwise, $\pi^{\star}(X_i) = 1$.

## C. Datasets

### C.1. For Section 3.1

We use the dataset of (CDC, 2015). It is a tabular dataset, where each row corresponds to an individual; the targets $Y_i$ in the original dataset denote whether the individual was (i) diabetic, (ii) pre-diabetic, or (iii) neither. For the purposes of our experiment, we only look for whether they were diabetic or not. The covariates are effectfully responses to the following simple survey questions:

- "do you have high blood pressure?"

- "do you have high cholesterol?"

- "how long has it been since the last time you have checked your cholesterol levels?"

- "what is your body mass index (BMI)?"

- "have you smoked at least 100 cigarettes in your entire life?"

- "has you ever been told you had a stroke?"

- "have you been diagnosed with coronary heart disease (CHD) or myocardial infarction (MI)?"

- "how much physical activity have you done in the past 30 days (excluding job)?"

- "how often do you consume fruit?"

- "how often do you consume vegetables?"

- "how often do you consume alcohol?"

- "do you have health care coverage, including health insurance, prepaid plans such as HMO, etc.?"

- "Was there a time in the past 12 months when you needed to see a doctor but could not because of cost?"

- "Would you say that in general your health is: [excellent / very good / good / fair / poor]"

- "Now thinking about your mental health, which includes stress, depression, and problems with emotions, for how many days during the past 30 days was your mental health not good?"

- "Now thinking about your physical health, which includes physical illness and injury, for how many days during the pat 30 days was your physical health not good?"

- "Do you have serious difficulty walking or climbing stairs?"

- "What is your age?"

- "What is your highest level of education?"

- "What is your level of income?"

### C.2. For Sections 3.2 and 3.3

We use the dataset of (Blackard, 1998). Upon this dataset, in a training split, we train a simple random forest classification model. We also separate a validation split to compute the validation loss in Section 3.2. At evaluation time:

- For the 'non-poisoned' data stream in Section 3.2, where the null should *not* be rejected, we just use the data remaining after the training and validation splits.

- For the 'poisoned' data stream in Section 3.2, we switch the label with a probability of

$$\text{clamp}_{[0,1]}\left(\left(\frac{t}{0.5}\right)^2\right),$$

  for time $t \in [0, 1]$.

- For the data stream in Section 3.3, we switch the label with a probability of

$$\mathbb{1}[t \geq 0.23] \cdot \text{clamp}_{[0,1]}\left(\left(\frac{t+1}{5} + 0.35\right)^2\right),$$

  for time $t \in [0, 1]$. The indicator causes a visible change in the time series, good for visualization. The remaining bit is done differently from in the previous section so that the change in the distribution is not too drastic.

### C.3. For Section 3.4

We generate a random DAG with 6 nodes using the Erdös-Renyi procedure, and mark the last three of these nodes as 'costly'. Relations between the nodes are given by linear functions, whose weights and biases are sampled randomly, with additional independence gaussian noise with a standard deviation of 0.4.

# D. Code

The source code to reproduce the experiments in the paper, as well as additional experiments (e.g. varying seed, varying sampling budget, underpowered settings [i.e., with worse predictive models]), is available at `https://github.com/dccsillag/experiments-prediction-powered-evalues`.

