# OpenReview forum: "Prediction-Powered E-Values"
_ICML.cc/2025/Conference — ICML 2025 poster_

### Official Review · Reviewer_piUd · 2025-03-08

**Overall Recommendation:** 2

**Summary:**

The authors propose to combine e-values to prediction-powered inference. The result is an e-value that combines in its definition both observed data as well as prediction of an auxiliary model. This approach is shown to improve the power of the testing procedure as long as the model is good enough. The author then provide three example applications encompassing mean estimation, change point detection and casual discovery. In all these cases, the PPI e-values are have better statistical power.

**Claims And Evidence:**

The claims follows straightforwardly from the properties of e-value and PPI, and they are correct and clearly stated. The authors also do a good job of at explaining the implications of the theoretical results.

**Essential References Not Discussed:**

Active, anytime-valid risk controlling prediction sets from Ziyu Xu, Nikos Karampatziakis and Paul Mineiro in NeurIPS 2024.

**Experimental Designs Or Analyses:**

Yes and they seem all correct.

**Methods And Evaluation Criteria:**

Yes. All the considered evaluation criteria are well-suited to measure the statistical power of the testing procedure.

**Other Comments Or Suggestions:**

My main suggestion is to include and clarify the similarities and difference from prior art.

##update after rebuttal: I decided to keep my score considering the overlap with existing work "Active, Anytime-Valid Risk-Controlling Prediction Sets".

**Other Strengths And Weaknesses:**

I think the paper is very well-written, and the author does a great job of providing theoretical claims along with their practical implications. The idea of combining e-values and PPI is valid, and I appreciate the analysis of the growth rate of the proposed scheme. The experiments are well-designed, significant, and effectively support the paper’s claims.

From my perspective, the main weakness of the paper is that the same idea has already been presented in Active, Anytime-Valid Risk-Controlling Prediction Sets by Ziyu Xu, Nikos Karampatziakis, and Paul Mineiro. See the section on "Variance reduction through prediction" in which the same e-processes based on PPI is proposed. This work was originally posted on ArXiv in June 2024, so I am unsure whether it qualifies as concurrent work. At the very least, it should be mentioned, and the differences between the two should be discussed.

**Questions For Authors:**

N/A

**Relation To Broader Scientific Literature:**

I think the paper misses a critical reference Active, anytime-valid risk controlling prediction sets from Ziyu Xu and Nikos Karampatziakis and Paul Mineiro in NeurIPS 2024. In this paper the authors " describe how to use predictors (i.e., the machine learning model for which we provide risk control guarantees) to further improve the utility of our RCPSes by estimating the expected risk conditioned on the covariates" which to me appear to very similar to the approach presented in the paper.

**Theoretical Claims:**

Yes, I have checked the theoretical claim about the growth rate and size of confidence intervals. They seem all correct.

---

> ### Author Rebuttal · Authors · 2025-03-31
>
> Thank you for pointing out the paper of [Xu et al.]. It is definitely a work we need to cite, and we will do so. That said, our contribution differs from theirs.
>
> 1. Our construction is significantly more general than theirs. In equation (8) they construct what would be our prediction-powered e-process in the specific case of the mean-estimation e-process of [Waudby-Smith and Ramdas, 2021]; this particular construction can alternatively be seen as the standard mean-estimation e-process applied to the expectation constructed in [Zrnic & Candès, 2024]. Our construction, in contrast, is valid for virtually any e-value, which is much broader. For example, it is not at all clear how their construction could be leveraged for causal discovery, whereas ours is applicable in a straight-forward manner.
> 2. The context is fairly different. In their paper, the primary concern is the construction of predictive sets that are endowed with a strong notion of validity. In our paper, we are concerned with statistical inference.
> 3. Beyond constructing the e-values, we also investigate the statistical power of our procedure (growth rate, etc.) in the general case, which is something that [Xu et al.] do only in their restricted setting and for a lower bound on the growth rate (not the exact growth rate, as we do). Also note that our analyses on the power are not exactly trivial, as evidenced by the fact that stronger (but less compact) statements are left to the supplementary material.
>
> Overall, given these differences, we believe that our work is a significant contribution, and we hope the reviewer agrees.

---

> > ### Comment · Reviewer_piUd · 2025-04-03
> >
> > Thank you for considering my comments. I agree with the points discussed above, and I believe that, in light of our discussion, the contribution of the paper would be best restated as extending and refining the ideas presented in Xu et al. Would the authors agree with this perspective?

---

> > > ### Author Response · Authors · 2025-04-08
> > >
> > > Thank you for your response. We agree that our results can be seen as a more general framework from which the work of [Xu et al.] can be derived as a special case. This connection certainly justifies the citation of their work, which we will add. Thank you for bringing this interesting paper to our attention.

---

### Official Review · Reviewer_gWwp · 2025-03-11

**Overall Recommendation:** 4

**Summary:**

This paper introduces prediction-powered e-values. Its primary contribution is extending prediction-powered inference (PPI) beyond Z-estimation problems (e.g. inference of means) to the broader class of inference problems solvable via e-values. The authors show that their method retains key benefits of classic e-values, such as anytime-validity. Moreover, unlike classic PPI, their method allows updating the predictive model throughout the inference process. Empirical validation demonstrates its effectiveness in diverse scenarios, consistently yielding more precise inference.

**Claims And Evidence:**

The theoretical claims are convincingly supported.

The empirical evidence could be improved; see Methods And Evaluation Criteria.

**Essential References Not Discussed:**

I believe there are essential related works missing from the current submission. In particular, the literature on semiparametric inference and missing data is highly relevant but currently omitted.

The prediction-powered e-value construction (lines 148-156) strongly resembles doubly robust estimators, introduced by:

- J. M. Robins, A. Rotnitzky, and L. P. Zhao, *“Estimation of regression coefficients when some regressors are not always observed,”* Journal of the American Statistical Association, 89(427), pp. 846–866, 1994.
- J. M. Robins and A. Rotnitzky, *“Semiparametric efficiency in multivariate regression models with missing data,”* Journal of the American Statistical Association, 90(429), pp. 122–129, 1995.

The main conceptual difference in the "modern" PPI framework is that the imputation model is trained externally on large independent datasets (see e.g. Section 5 in [1] and Appendix A.2 in [2] for a discussion). However, in this submission, the authors train the imputation model on the available labeled data, making their approach essentially an extension of classic semiparametric inference into a sequential setting with e-values.

Explicitly discussing these connections would clarify the paper's position in relation to the established literature.

[1] Xu, Zichun, Daniela Witten, and Ali Shojaie. "A Unified Framework for Semiparametrically Efficient Semi-Supervised Learning." arXiv preprint arXiv:2502.17741 (2025).

[2] De Bartolomeis, P., Abad, J., Wang, G., Donhauser, K., Duch, R. M., Yang, F., & Dahabreh, I. J. (2025). Efficient Randomized Experiments Using Foundation Models. arXiv preprint arXiv:2502.04262.

**Experimental Designs Or Analyses:**

N/A

**Methods And Evaluation Criteria:**

Some aspects of the experimental setup could be improved:

- Choice of labeling proportion (Sections 3.1 and 3.2):
  The labeling budgets chosen (1% and 0.5%) are overly restrictive and somewhat arbitrary. An ablation study varying these proportions would improve the understanding of the method's effectiveness across different budget constraints.

- Qualitative nature of causal discovery evaluation (Section 3.4):
  The causal graph experiments currently rely on a single visualized random graph, making conclusions overly qualitative. To properly validate their method, the authors could sample multiple random graphs systematically and provide quantitative metrics (e.g. structural Hamming distance) averaged across these graphs.

**Other Comments Or Suggestions:**

- typo line 87: ... over the distribution of the Y∗.s

**Other Strengths And Weaknesses:**

- I believe the algorithm is not described in sufficient detail. Specifically, the procedure for estimating and sequentially updating the predictive model is not clearly discussed, despite being central to the performance of the method. Providing details on how these models are initially trained and then updated (e.g., online vs. batch updates, frequency of updates) would be essential for understanding and replicating the approach.
- An important limitation not explicitly discussed by the authors is that if the predictive model yields very inaccurate predictions (e.g. misspecified model), their proposed method could actually result in worse statistical efficiency or lower power compared to simply using the labeled data alone. A brief discussion or exploration of this potential issue—either theoretically or empirically—would strengthen the paper.

**Questions For Authors:**

- Your prediction-powered e-value appears structurally very similar to doubly robust estimators. Can you explicitly clarify how your approach differs from (or relates to) these classic doubly robust/AIPW methods, particularly since you train the predictive (imputation) models online using labeled samples rather than externally obtained data?

- You chose labeling budgets of 1% (diabetes prevalence estimation) and 0.5% (risk monitoring). What motivated these particular choices, and how sensitive are your empirical conclusions to these specific labeling budgets? Could you provide some evidence of robustness across different budget values?

- You currently provide causal discovery results on a single random DAG. Could you run your method across multiple random graphs and provide quantitative performance metrics (e.g., structural Hamming distance)?

**Relation To Broader Scientific Literature:**

See: Essential References Not Discussed

**Theoretical Claims:**

I did not check the correctness of any proof.

---

> ### Author Rebuttal · Authors · 2025-03-31
>
> We'd like to thank the reviewer for their review. Please refer to our responses below.
>
> **Labeling budgets:** We agree. The choices of 1% and 0.5% are indeed a bit arbitrary, and were made mainly so as to be a reasonably large reduction in the number of collected labels, while keeping the underlying number of samples reasonably large. We will add experiments varying the budget in the camera-ready version. That said, the currently presented results aren't particularly sensitive to these sampling budgets, though if the number of labelled samples (not the proportion, the absolute number) goes under a certain threshold, then our method does struggle a bit (most likely because the predictive model deteriorates).
>
> **Evaluation of causal discovery setting:** The reviewer raises a good point. We will add comparisons of the average structural Hamming distance over several sampled random graphs to the camera-ready version. We have already run some first simulations, with favorable results: over 40 sampled graphs, the baseline of only labelled graphs obtains a mean structural Hamming distance of 12.15, whereas our method obtains that of 6.7; the best possible one (i.e., if we used all data available, as in Figure 4) is of 6.4.
>
> **Connections to the semiparametric inference and missing data literature:** Indeed, our prediction-powered e-values are essentially an AIPW-like estimator applied to the e-values. This construction was inspired from [Zrnic & Candès, 2024], which were in turn inspired by the usual doubly-robust estimators literature (cf. right after their equation (1)). We will make this connection more explicit in our paper. That said, perhaps the key difference between our method and usual semiparametric inference is that, rather than applying the AIPW-like estimator to the data, we apply them to the e-values -- i.e., directly to our measure of evidence. By doing so, we avoid asymptotics, and inherit all the properties of the original e-values, since all we require is that the e-value be at most one in expectation. The ability to update the predictor sequentially comes for free and is fairly easy to prove, with no significant restrictions (e.g. regularity conditions) necessary.
> It is not clear whether our method can be seen as a more classic doubly-robust estimation for general e-values (as [Xu et al.] and [De Bartolomeis et al.] do for some previous PPI procedures). In the particular case of the e-value for the mean [Waudby-Smith and Ramdas, 2021], one can look at the resulting PPI e-value as the e-value for the mean applied on the AIPW-like estimator of [Zrnic & Candès]. But this observation does not seem to generalize to more complex e-values (e.g., those of https://arxiv.org/abs/2305.00143, or ones arising from p-to-e calibration).
>
> **Details on the use of the algorithm in the experiments:** In the experiments, the sequential updating of the predictive model was done by simply appending new available data and retraining on all (labelled) data prior, every 100 collected samples. We will add this to the experiments section.
>
> **Discussion on cases where PPI e-values would be underpowered:** Thank you for the suggestion, we will add this.

---

### Official Review · Reviewer_t47N · 2025-03-13

**Overall Recommendation:** 4

**Summary:**

This paper proposes an extension of "prediction powered inference" to e-values, a general class of statistics that are used in many methods outside of the estimation. The authors show theoretical results justifying the validity of their prediction powered e-value, and its validity under active sampling, as well results for its growth rate. They also provide a comprehensive set of experiments showing the applications of prediction powered e-values to change point detection, continuous monitoring, and causal discovery.

**Claims And Evidence:**

Yes.

**Essential References Not Discussed:**

None

**Experimental Designs Or Analyses:**

Yes, all 4 experiments. No particular issues

**Methods And Evaluation Criteria:**

Yes.

**Other Comments Or Suggestions:**

- Is it possible to tie the growth rate result in Theorem 2.2 to a property of the predictor, e.g., MSE?
- One potential drawback is that if one's prediction are quite off, the resulting PPI e-value can be much smaller than the original true e-value --- is there a way of mitigating this and falling back to sampling every true e-value if the predictor is bad?
- A nitpick, but the original prediction powered inference reduced variance of the estimator based on the accuracy of the predictor, even without active sampling. For prediction powered e-values, it seems like it's advantage lies solely in the active setting, and does not offer any benefits in the "semi-supervised" setting where a fixed batch of true e-values is collected. Is there any way to also use prediction powered e-values in the "semi-supervised" setting?

**Other Strengths And Weaknesses:**

The experiments are quite substantial and illustrate a suit of real world use cases, which is nice.

**Questions For Authors:**

None.

**Relation To Broader Scientific Literature:**

This paper extends active prediction powered inference methods to work with e-values. There has been similar work (mathematically) in the e-value literature in the past for off policy estimation/doubly robust estimation/active risk control, but never applied in a black-box fashion to e-values directly.

**Theoretical Claims:**

Yes --- specifically for Theorem 2.1 --- seems correct.

---

> ### Author Rebuttal · Authors · 2025-03-31
>
> We'd like to thank the reviewer for their insightful comments.
>
> **Growth rate as a function of the MAE of the predictor:** Yes, we can; in particular, we can show that the Wasserstein distance in Theorem 2.2 is upper bounded by the MAE of the predictor:
> $$\begin{aligned}
>     W(\mu\_i(X\_i) || Y\_i)
>     &= \sup_{\\|\phi\\|\_{Lip} \leq 1} | \mathbb{E}[\phi(\mu\_i(X\_i))] - \mathbb{E}[\phi(Y\_i)] |
>     \\\\ &= \sup_{\\|\phi\\|\_{Lip} \leq 1} | \mathbb{E}[\phi(\mu_i(X\_i)) - \phi(Y\_i)] |
>     \\\\ &\leq \sup_{\\|\phi\\|\_{Lip} \leq 1} \mathbb{E}[| \phi(\mu\_i(X\_i)) - \phi(Y\_i) |]
>     \\\\ &\leq \sup_{\\|\phi\\|\_{Lip} \leq 1} \mathbb{E}[| \mu\_i(X\_i) - Y\_i |]
>     \\\\ &= \mathbb{E}[| \mu\_i(X\_i) - Y\_i |].
> \end{aligned}$$
> This is an interesting result, and we will add it to the camera-ready version of the paper. Thank you for the suggestion!
>
> **Fallback to non-PPI e-value in case of lack of power:** There probably does not exist a procedure which allows for the practitioner to 'peek' at the final prediction-powered e-value $E^{ppi}_n$ and non-prediction-powered one $E_n$, and choose which to use. However, if at any point over the course of the inference the practitioner believes that the prediction-powered e-values are under-powered (e.g., because so far not using it would have been more efficient), then can start selecting $\pi_i(X_i) = 1$, which would reduce the prediction-powered e-value to the non-prediction-powered one.
> Alternatively, since we are working with e-values, any convex combination (parametrised by some scalar $\eta_i \in [0, 1]$) of the e-value components $\eta_i e^{ppi}_i + (1 - \eta_i) e_i(Y_i)$ yield a valid e-value, as long as the $\eta_i$ are predictable (and thus independent of $e^{ppi}_i$ and $e_i(Y_i)$). So rather than increasing the probability of sampling the label, one could also decrease such a $\eta_i$.
>
> **Semi-supervised setting:** Indeed, our construction is focused on an active setting, and extending it to a "semi-supervised" setting is non-trivial. Nevertheless, here is one possible way of doing it (though probably suboptimal): at each step, choose $\pi_i$ to approximate the probability of whether you have access to the corresponding label. Sample $\xi_i\sim Bern(\pi_i(X_i))$. Then you have a couple of cases:
>
> 1. If $\xi_i = 0$ (i.e., don't want to use the label), we continue as normal.
> 2. If $\xi_i = 1$ (i.e., want to use the label) and we have access to the label, we continue as normal.
> 3. If $\xi_i = 1$ (i.e., want to use the label) but we _don't_ have access to the label, then rather than contributing $e^{ppi}_i$ to the e-value, we contribute $e^{ppi-missing}_i:=\frac{a_i-(1-\pi_i(X_i))b_i}{\pi_i(X_i)}$; this works because, whichever the value of $Y_i$, $e^{ppi-missing}_i \leq e^{ppi}_i$.
>
> We believe that a more in-depth exploration of "semi-supervised" procedures such as this one is best suited to its own paper (e.g., how do we retain good performance even when case (3.) is somewhat frequent?), and so we leave it to future work.

---

### Official Review · Reviewer_L9yP · 2025-03-14

**Overall Recommendation:** 4

**Summary:**

The paper proposes a methodology for converting any e-value based inference procedure into a prediction-powered counterpart. More concretely, they consider the setting where we have a data stream $(X_i, Y_i, \pi_i, \xi_i)_{i=1}^\infty$ where $X_i$ is cheap data, $Y_i$ is expensive labelled data, and $\xi_i \sim Bern(\pi_i(X_i))$ indicates whether we have access to $X_i$ or not. Additionally, we have access to a predictive model $\mu_i$ at each step.

In this model, they key contribution is that given any specification of a sequence of e-values on the expensive data, $E_n \doteq \prod_{i=1}^n e_i(Y_i)$, they show one can construct a corresponding prediction-powered sequence $E_n^{ppi} \doteq \prod_{i=1}^n e_i^{ppi}(Y_i)$ by defining $e_i^{ppi} \doteq e_i (\mu_i(X_i)) + [e_i(Y_i) - e_i(\mu_i(X_i))] \cdot \frac{\xi_i}{\pi_i(X_i)}$.

The authors supplement this contribution with a lower bound on the expected growth rate that is in terms of that using the true data minus the expected Wasserstein distance between the predictions of the model and the true values $Y_i$. They extend their results to building anytime-valid prediction-powered confidence sequences, as well as to general e-value based algorithms.

Finally, they supplement their theoretical contributions by a series of experiments showing the empirical gains achievable (faster null rejection, smaller interval widths, etc.).

**Claims And Evidence:**

Yes.

**Essential References Not Discussed:**

Not to my knowledge.

**Experimental Designs Or Analyses:**

I found the experimental analyses to be thorough and well-done. I appreciated the inclusion of different types of inference tasks, the baseline considered and the presentation.

**Methods And Evaluation Criteria:**

Yes.

**Other Comments Or Suggestions:**

List of typos:
- on line 57, left column, i think it should be “allow” rather than “allows”
- on line 175, right column, change “inputted” to “imputed”
- on line 225, left column, it should be “thorough” not “through”
- on line 357, right column, “extremely quicker” is not grammatically correct

**Other Strengths And Weaknesses:**

Both prediction-powered inference and inference with e-values are relatively recent developments in statistical inference that are particularly appealing and relevant — proposing an elegant way to bring them together is in my opinion a very useful contribution.

Beyond allowing for prediction-powered inference with e-values, this methodology allows for continuous updating of the predictive model used during inference, which I find to be particularly neat and practical (as batch predicition-powered inference required a predictive model fixed a priori).

One potential weakness is that the modeling of missingness as $Bern(\pi_i(X_i))$ is a bit weird and a bit hard to map to all practical scenarios where one may want to employ prediction-powered inference. For example, sampling $\xi \sim Bern(\pi_i(X_i))$ may not be possible and/or unclear if we could have access to $\pi_i(X_i)$ in order to construct our e-value in many applications.

**Questions For Authors:**

Could you comment more on modeling missingness through $\pi_i(X_i)$?

**Relation To Broader Scientific Literature:**

This paper is very fundamentally connected to the broader scientific literatures of sequential hypothesis testing via betting and the more recent line of work in prediction-powered inference. They provide extensive references and even apply their method to more niche/particular developments such as using e-values for change-point detection. In my opinion, this is a situation where combining two lines of work really enriches both respective lines.

**Theoretical Claims:**

I skimmed through the proofs in the appendix and they appear correct.

---

> ### Author Rebuttal · Authors · 2025-03-31
>
> We'd like to thank the reviewer for their comments.
>
> **Regarding modelling missingness through $\pi_i(X_i)$:** Our construction is focused on an active/experimental setting in which we can actually sample $\xi_i \sim Bern(\pi_i(X_i))$ and collect $Y_i$ when $\xi_i=1$. Extending it to a "semi-supervised" setting is non-trivial. Nevertheless, here is one possible way of doing it (though probably suboptimal): at each step, choose $\pi_i$ to approximate the probability of whether you have access to the corresponding label (this is akin to e.g. a propensity score). Sample $\xi_i\sim Bern(\pi_i(X_i))$. Then you have a couple of cases:
>
> 1. If $\xi_i = 0$ (i.e., don't want to use the label), we continue as normal.
> 2. If $\xi_i = 1$ (i.e., want to use the label) and we have access to the label, we continue as normal.
> 3. If $\xi_i = 1$ (i.e., want to use the label) but we _don't_ have access to the label, then rather than contributing $e^{ppi}_i$ to the e-value, we contribute $e^{ppi-missing}_i:=\frac{a_i-(1-\pi_i(X_i))b_i}{\pi_i(X_i)}$; this works because, whichever the value of $Y_i$, $e^{ppi-missing}_i \leq e^{ppi}_i$.
>
> We believe that a more in-depth exploration of "semi-supervised" (i.e., that rely solely on observational data) procedures such as this one is best suited to its own paper (e.g., how do we retain good performance even when case (3.) is somewhat frequent?), and so we leave it to future work.
>
> Also, thank you for pointing out the typos, we will correct them!

---

### Official Review · Reviewer_3dJo · 2025-03-14

**Overall Recommendation:** 3

**Summary:**

This paper extends the ideas of prediction-powered inference to e-values. The contribution is focused but nice, since as the authors note, prediction-powered e-values allow for a broader set of possible inference techniques and guarantees (anytime validity in particular). The core methodology involves leveraging the same inverse propensity weighting tricks used in AIPW and Active Statistical inference to the sequence of components of an e-value (with additional constraints on $\pi(X_i)$ to ensure that $E_n^{ppi}$ is non-negative for all $n \in \mathbb{N}$), and then theoretically showing that the ppi e-value is a valid e-value. The remainder of the empirical section gives several compelling experiments that show the versatility and effectiveness of prediction-powered e-values.

**Claims And Evidence:**

The theoretical claims are all clear and well-presented. I appreciate the focus on crisp statements in the main text, with additional generalizations in the Appendix.

There are, however, two claims that I believe need more support:
- Contribution 3 cites "massive (often 100x-1000x) reductions in data acquisition costs", but from the empirical results it is not clear where this is supported. Please also see the "Questions for Authors" section of this review about the labeling and descriptions of the results figures.
- It is repeatedly mentioned that a strength of sequential prediction-powered e-values is the ability to update the underlying prediction model and data collection rule over the course of the inference process. However, for at least the experiment in 3.1, this is also possible in Active Statistical Inference (Zrnic & Candes, 2024), which relies only on having martingale increments (and uses the martingale CLT). It seems the fairest comparison would be to use the same setup used in "Active prediction-powered (ours)", but with the estimator in Zrnic & Candes. Similarly, it is unclear if the vanilla prediction-powered approach uses lambda-tuning or not?

**Essential References Not Discussed:**

The related work appears to be discussed appropriately throughout the introduction and main text.

**Experimental Designs Or Analyses:**

The experimental designs in all of the four case studies appear sound for demonstrating the applicability of the proposed method. As noted earlier, a comparison to Active Statistical Inference (Zrnic & Candes, 2024), which allows for updates to the predictive model and adaptive sampling, is missing from the experimental section.

**Methods And Evaluation Criteria:**

Yes.

**Other Comments Or Suggestions:**

See above.

**Other Strengths And Weaknesses:**

The main issue with statistical inference using e-values is typically a loss in power. Therefore I am surprised by the improvement over PPI in Fig. 1, though the authors claim this is due to data splitting vs. online model updates. The paper should be comparing on fairer terms to Active Statistical Inference (Zrnic & Candes, 2024) in Fig. 1.

**Questions For Authors:**

While the presentation is generally very good, a couple of things were unclear:

1) What is $m$ on page 5 for the definition of $E_n^{(\theta)}$?

2) What is $n$ in the experiments in Fig. 1? I understand that the number of labelled examples is 1% of that. Similarly, what $n$ is used for vanilla PPI, as that seemed to be different following data splitting.

3) What is the x axis in Fig. 2?

4) Can you provide an explanation of the change point detection algorithm in Shekhar & Ramdas in the Appendix.

**Relation To Broader Scientific Literature:**

The paper builds upon the existing literature on prediction-powered inference, particularly the work of Angelopoulos et al. (2023) and Zrnic & Candes (2024). It expands this area by applying the concept to e-values, which have gained popularity as a flexible alternative to p-values --- with impactful applications to sequential and post-hoc inference. The paper also connects to the literature on e-values in various applications like hypothesis testing, confidence sequences (Waudby-Smith & Ramdas, 2020), change-point detection (Shin et al., 2022; Shekhar & Ramdas, 2023), and causal discovery. The key contribution is in bridging these two active areas of research.

**Theoretical Claims:**

Core results (Theorem 2.1 & Proposition 2.3) were checked, and auxiliary results (Theorem 2.2, Proposition 2.3/2.4/2.6) appear to be correct.

---

> ### Author Rebuttal · Authors · 2025-03-31
>
> We thank the reviewer for their constructive remarks, addressed as follows:
>
> **Sequential updating and [Zrnic & Candès, 2024]:** The reviewer raised a good point. We will add comparisons in Section 3.1 to the method of [Zrnic & Candès, 2024]. We've run a preliminary experiment on this and indeed, their method (the sequential variant) performs much more similarly to ours (their martingale-CLT-based interval is slightly smaller than our nonasymptotic e-value-based one: with fixed label sampling probability, ours is [0.127, 0.155], while theirs is [0.128, 0.153]; for reference, vanilla PPI is [0.129, 0.172]).
>
> **Reductions in acquisition costs:** In all the experiments, we stipulate some particular labelling budget, in the form of $\pi_{inf}$. For example, in Section 3.1 we have $\pi_{inf} = 1\%$ (right under equation (2)) and in Section 3.2 we have $\pi_{inf} = 0.5\%$ (right after equation (4)). These quantities denote the percentage of samples which we will label, and it is from these numbers that we obtained the 100x-1000x statistic: $1 / 1\\\% = 100$, and $1 / 0.5\\\% = 200$. As for the 1000x number, it corresponded to a previous version of Section 3.2 that we ended up not including in the paper, in which we had $\pi_{inf} = 0.1\%$. The results were very much satisfactory, but we ended up increasing $\pi_{inf}$ so that it was a bit less arbitrary and more in-line with the other values of $\pi_{inf}$ in the paper. After this change, it seems we forgot to update the introduction. We will fix this for the camera-ready version. We will also clearly present these numbers in the experiments section, as well as compute the exact amount of labels used in practice (though the actual number was generally very close to the stipulated budget).
>
> **As for the remaining questions:**
> - $m$ on page 5: that is supposed to be $\theta$, we'll fix that.
> - $n$ in the experiments in Fig. 1: the full dataset has 253680 samples; for the only-labelled baseline, we use 253680*1% = 2537 of that, for our method we follow the sampling (which samples about 1% of labels as well), and for vanilla PPI we collect also 1% of labels, and split them 50%-50% between training the model and the statistical inference, since we both want to get a good model (so want a large training split) but also don't want to suffer much in the statistical inference (so want a large inference split). We will also include experiments varying this split's proportion, though it does not seem to improve vanilla PPI significantly.
> - The x axis in Fig. 2 is time, i.e., the course of the inference procedure. The first samples of the experiment are to the left, the later samples are to the right. We will make this more clear in the camera-ready version.
> - We will also add an explanation of the change-point detection algorithm of Shekhar&Ramdas in the Appendix, thank you for the suggestion.

---

### Decision · Program_Chairs · 2025-05-01

**Decision:**

Accept (poster)

**Comment:**

The reviewers appreciated the contribution of the paper that is a general framework that marries two timely topics: using prediction-power inference to make e-value calculations more sample-efficient/yielding higher power. The experiments are convincing and support the relevance of the work.

Please incorporate reviewer's comments in your final draft, e.g. add the connection to AIPW estimators in your positioning and further, make clear that rather than semi-supervised setting, you consider the active/experimental setting where you can obtain labels according to $\pi_i X(_i)$ rather than trying to model and learn $\pi$. However, similar to doubly-robust AIPW, my hunch would be that you can readily use those arguments to make it work (asymptotically) for the semi-supervised case.